# Synergism of Multi-Modal Data for Mapping Tree Species Distribution—A Case Study from a Mountainous Forest in Southwest China

**Pengfei Zheng** [1], **Panfei Fang** [1], **Leiguang Wang** [2,3], **Guanglong Ou** [1], **Weiheng Xu** [2,3], **Fei Dai** [2] **and Qinling Dai** [4,*]

1   Faculty of Forestry, Southwest Forestry University, Kunming 650024, China
2   Institute of Big Data and Artificial Intelligence, Southwest Forestry University, Kunming 650024, China
3   Key Laboratory of National Forestry and Grassland Administration on Forestry and Ecological Big Data, Southwest Forestry University, Kunming 650024, China
4   Art and Design College, Southwest Forestry University, Kunming 650024, China
*   Correspondence: daiqinling@126.com

**Abstract:** Accurately mapping tree species is crucial for forest management and conservation. Most previous studies relied on features derived from optical imagery, and digital elevation data and the potential of synthetic aperture radar (SAR) imagery and other environmental factors have, generally, been underexplored. Therefore, the aim of this study is to evaluate the potential of fusing freely available multi-modal data for accurately mapping tree species. Sentinel-2, Sentinel-1, and various environmental datasets over a large mountainous forest in Southwest China were obtained and analyzed using Google Earth Engine (GEE). Seven data cases considering the individual or joint performance of different features, and four additional cases considering a novel clustering-based feature selection method, were analyzed. All 11 cases were assessed using three machine learning algorithms, including random forest (RF), support vector machine (SVM), and extreme gradient boosting tree (XGBoost). The best performance, with an overall accuracy of 77.98%, was attained from the case with all features and the random forest classifier. Sentinel-2 data alone exhibited similar performance as environmental data in terms of overall accuracy. Similar species, such as oak and birch, cannot be spectrally discriminated based on Sentinel-2-based features alone. The addition of SAR features improved discrimination, especially when distinguishing between some coniferous and deciduous species, but also decreased accuracy for oak. The analysis based on different data cases and feature importance rankings indicated that environmental features are important. The random forest outperformed other models, and a better prediction was achieved for planted tree species compared to that for the natural forest. These results suggest that accurately mapping tree species over large mountainous areas is feasible with freely accessible multi-modal data, especially when considering environmental factors.

**Keywords:** tree species mapping; mountainous forest; Sentinel-1 and Sentinel-2; environmental data





## 1. Introduction

As an essential component of ecosystems, forest vegetation plays a vital role in regulating climate change, monitoring biodiversity, estimating carbon sequestration, and promoting sustainable forest operations [1–3]. Acquiring accurate and low-latency information on tree species is crucial for the ability of forest authorities to implement effective forest management and monitor biodiversity. However, the species diversity conditions resulting from the complex composition of vegetation types, dense forest coverage and typical climatic characteristics present a challenge when mapping tree species in mountainous forests. In addition, acquiring accurate sample data is challenging in these forests, where access is hindered by a lack of infrastructure and rugged terrain [4].

Remote sensing analyses are less costly in terms of labor and time than field surveys and aerial photography and are easily extended to large scales. The Landsat and Sentinel-2 series satellites provide images with high spatial resolutions, and the accumulation of these data with available revisit cycles can facilitate the mapping of forest types and changes in the distributions of specific tree species. As remote sensing technologies have developed, satellite data with different spectral, temporal, and spatial resolutions provided more chances for tree species mapping. Many recent studies have explored high-resolution remote sensing imagery such as IKONOS, QuickBird, and WorldView imagery to map forest types in detail, even at the species level [5–7]. Despite the advantages of these data in terms of the spatial details they provide, previous studies have been limited to relatively small study areas of a few hundred to a few thousand square kilometers. Thus, these data can only meet the needs of specific users due to their high user costs and time-consuming characteristics and cannot support forest inventories over extensive areas [8]. Therefore, medium- and high-resolution satellite data are more suitable for mapping tree species over widespread areas.

Sentinel-2 data are widely employed for classifying land use, forest types, and tree species [9,10]. Moreover, Sentinel-2 data contain red-edge bands, which are crucial spectral wavelength domains and can help discriminate the subtle differences among morphologically similar tree species [11]. Multitemporal or time-series data can capture subtle vegetation changes in phenology [12–15]. Using the dense time-series data derived from the Sentinel-2 satellite over the growing season, researchers can obtain detailed information regarding tree species' spectral–temporal patterns [16]. However, clouds can lead to invalid observations in some periods, especially in the rainy season of subtropical and tropical regions. Recent studies have used time-series smoothing and interpolation methods to preserve complete phenology information. Effective methods for processing image-based temporal features include fitting time-series curves and removing high-frequency noises with effective filters to generate gap-free time-series products [17]. Nevertheless, a significant number of available observations are needed to construct timeseries that describe species growth to ensure the accuracy of the output phenology features.

An active remote sensing synthetic aperture radar (SAR) that is not blocked by cloud cover or insufficient lighting can be used as a supplementary data source to overcome the limitations of optical satellites that are susceptible to cloudy and rainy weather [18]. SAR data can be used for crop identification, farmland parameter extraction, yield estimation, average forest stand height mapping, aboveground biomass estimation, forest condition change detection, and forest type classification tasks [8,19–21]. SAR systems are sensitive to the biochemical structure of vegetation and the dynamic characteristics of plant targets, such as the plant water content, geometric properties, and surface roughness; for forests, backscatter is influenced by the roughness of leafy branches and the morphology and orientation of the leaves [22,23]. Many studies have combined optical and radar sensors' strengths when capturing vegetation biochemical and physical properties for various applications in forests. For example, SAR data improved the ability to predict forest heterogeneity indices in mapping the diversity indices of forest plants [24]. The combination of Sentinel-1 with Sentinel-2 data allowed for the effective assessment of spatial vegetation heterogeneity and diversity over a wide area. The results of another tree species classification study suggested that adding the VV, VH, and ratio of both (VH/VV) features of SAR data based on Sentinel-2 data could improve the classification accuracy of coniferous forests [25]. In addition, seasonal SAR features have been reported to be helpful for forest area estimations and forest type classifications in temperate forests [26]. Therefore, combinations of Sentinel-1 and Sentinel-2 data can be used to improve the classification results for forests, especially in large subtropical mountainous regions where the availability of optical images is limited. In previous studies, topographic factors could improve the accuracy of forest type and tree species classifications, especially in mountainous regions [27]. The spatial distribution of tree species is greatly influenced by topography, which affects various environmental conditions, such as solar radiation, temperature, and moisture [28]. In addition

to topography, abiotic conditions such as precipitation, temperature and soil conditions also significantly affect the distributions of tree species and forests. These ancillary datasets can thus provide complementary information for discriminating among different tree species. Furthermore, several previous studies confirmed the strong correlations between climatic factors and tree species distributions and identified precipitation and temperature as the most critical factors influencing plant communities [29,30].

In contrast to optical and radar data, the spatial distribution of climate variables characterizes species' climate preferences but cannot directly measure vegetation; interestingly, such datasets have rarely been used for tree species classifications [31]. In tropical and subtropical forests, seasonal rainfall variations are highly apparent, and species drought performance and physiological drought tolerance positively impact species distributions [32]. In research assessing the importance of rainfall temperature and its seasonality on the distribution of tropical forest tree species, researchers found that 95% of the distributions of 20 species were significantly correlated with annual precipitation [30]. Studies often combine ancillary predictor variables with remotely sensed data, such as environmental data, to improve species classification accuracy. Combining multi-modal data sources leads to better results better than using remote sensing or ancillary environmental data alone; additionally, data source combinations can reduce the constraints of heterogeneous environmental conditions. To assess the performance of different source data, these data were iteratively added onto the single data to construct different feature sets, and these feature sets were used as inputs to different classifiers in a previous study [33]. Many studies have compared single or multiple data combinations to obtain the most suitable combination of the multi-source dataset [8,25,34]. Remote sensing-based tree species mapping tasks present an increasing demand for statistical methods. In the past, typical methods built on parametric analyses, including maximum likelihood, Bayesian, and some unsupervised clustering methods (e.g., the k-means), were employed [35–37]. Recently, some nonparametric machine learning methods, such as random forest (RF) and support vector machine (SVM) methods, have been widely used for tree mapping due to their stable performance when processing high dimensional features [38,39]. In addition, the proliferation of high-performance computing systems and data availability has increased and thus improved large-scale geospatial data processing capabilities. Google Earth Engine (GEE) is a cloud-based platform widely used for large-scale environmental monitoring and analyses; this platform can facilitate vegetation mapping at large regional scales [40].

The main objectives of this study are as follows:(1) to map nine tree species classes at a spatial resolution of 10 m in a typical tropical and subtropical climatic mountainous area, (2) to explore the potential of fusing multi-modal data for tree species classification in a mountainous area by combining Sentinel-1 backscatter, Sentinel-2 spectral, texture, time-series feature, and environmental variable data, and (3) to explore the optimal feature combination for tree species classification.

## 2. Materials and Methods

### 2.1. Study Area

Our study area is in the southwestern part of China, covering three administrative states of Yunnan Province, Lincang, Xishuangbanna, and Puer, spanning an area of over 70 thousand square kilometers (Figure 1). With 70% of the area covered by forests, the study area is one of Yunnan's most densely forested regions. The area has influence of the tropical and subtropical monsoon climate. It has distinct wet and dry seasons, with an average annual temperature of approximately 17.4 °C and an average annual precipitation of approximately 1489 mm. The topography of the study area is complex, consisting mainly of low- and medium-elevation mountains, foothills, and river valleys, ranging in elevation from 317–3429 m, and the total elevation difference is nearly 2000 m, with prominent vertical climate characteristics. The vegetation types in the study area belong to the southern Yunnan flora that evolved with the extrusion of the Indochinese landmass into Southeast Asia, influenced by the tropical Asian component since the end of the Tertiary era.

The vegetation in the study area is related closely to the Indo-Malaysian flora [41], resulting in a complex and diverse vegetation landscape including tropical rainforests, subtropical evergreen broadleaf forests, and subtropical mixed coniferous broadleaf forests. Our study area includes a significant amount of virgin forest. In recent years, the proportion of forest accounted for by plantation forests also increased quickly due to extensive economic crop cultivation and the policy of returning farmland to forest. The region is dominated by Simao pine (*Pinus kesiya var. langbianensis*), rubber (*Hevea brasiliensis*), eucalyptus (*Eucalyptus robusta*), Yunnan pine (*Pinus yunnanensis*), oak (Quercus L.), fir (*Cunninghamia lanceolata* (Lamb.) Hook.), birch (*Betula*), and alder (*Alnus cremastogyne* Burk.).

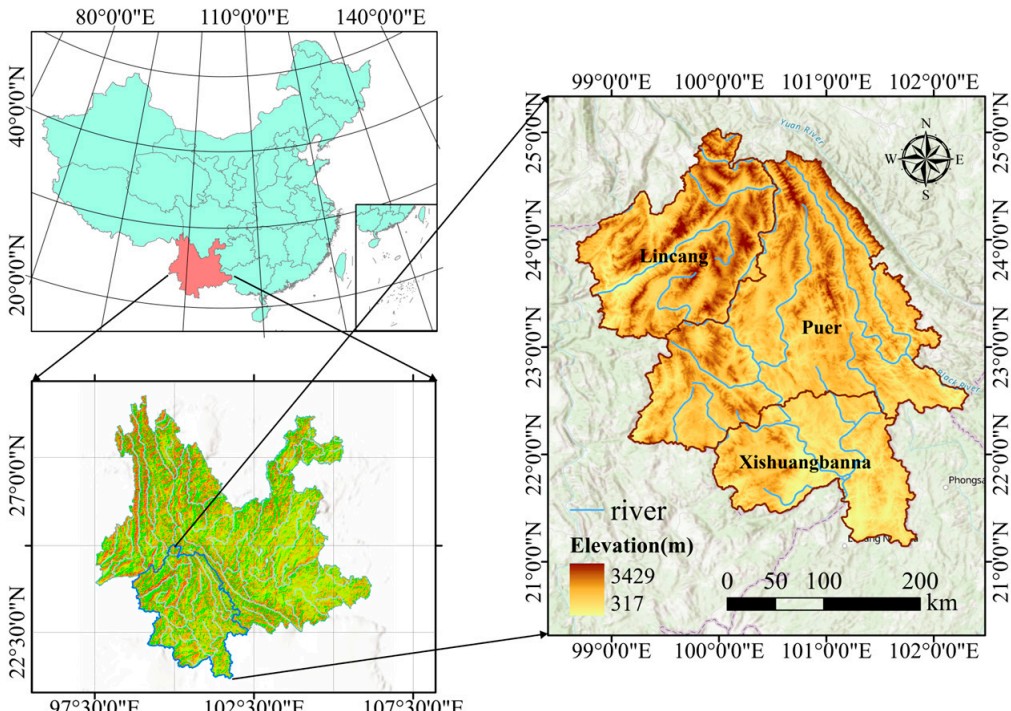

**Figure 1.** Location of the study area.

*2.2. Data and Preprocessing*

2.2.1. Features from Multi-Modal Data

Both Sentinel-2 A/B satellites carry multispectral instruments containing 13 bands, with spatial resolutions ranging from 10 to 60 m in the visible to shortwave infrared (SWIR) region. Sentinel-2 data were used in this study to generate three types of features: (1) spectral features, (2) 10-day time-series data, and (3) texture metrics. In the mountainous regions of Yunnan, it is challenging to obtain complete and appropriate images due to the presence of clouds, and cloud shadows, especially in cloudy and rainy summer months. To minimize the effects of missing pixels after cloud masking, we used Sentinel-2A/B multispectral top-of-atmosphere (TOA) reflectance images (Level-1C) from 2015 to 2017 to create the best composite pixels. Pixels obscured or covered by clouds, cloud shadows or snow were removed using a bitmask band (QA60) with cloud mask information in GEE. Furthermore, to fill the gaps that were present after the cloud-removal process, a median imagery composite was conducted with all observations from 2015 to 2017. Two types of spectral features were used to distinguish the tree species. The first type was the spectral bands of Sentinel-2, including blue, green, red, red-edge 1, red-edge 2, red-edge 3, Near-infrared (NIR), red-edge 4, SWIR1, and SWIR2 bands. In addition, 16 commonly used spectral indices were obtained: triangular vegetation index (TVI), land surface water index (LSWI),normalized difference water index (NDWI), normalized difference built-up index (NDBI), normalized burn ratio (NBR), red edge chlorophyll index (CRE), normalized difference salinity index (NDSI), normalized difference temperature index (NDTI), red-

edge normalized difference vegetation index (REDNDVI), red-edge position index (REP), normalized difference red edge index (NDRE), modified chlorophyll absorption ratio index (MCARI), medium-resolution imaging spectrometer terrestrial chlorophyll index (MTCI), inverted red-edge chlorophyll index (IRECI), normalized difference vegetation index (NDVI), and normalized difference senescent vegetation index (NDSVI). The crown shape can be used to characterize the age, growing conditions, and interspecies competition of trees of different species. The GEE cloud platform also provides the glcmTexture function for calculating a grey-level cooccurrence matrix (GLCM). Six parameters were calculated for red-edge 1 considering the referenced research results: the sum average, correlation, dissimilarity, variance, contrast, and cluster shade.

The red-edge band of Sentinel-2 is sensitive to chlorophyll and effectively separable across various plant phenology stages; the information in the red-edge band has been found to be critical in distinguishing tree species and the growth stages of species [42]. Therefore, enhanced REP vegetation index timeseries was produced to extract the phenological differences among tree species. Sentinel-2 did not provide efficient observations of the entire study area in 2016, so all observations from 2015 to 2017 were collected to perform 10-day median compositing. To resolve the issue of missing pixels after cloud masking, a time window interpolation algorithm was proposed by Park and Tateishi in 1998 [43]. Furthermore, the Savitzky-Golay (SG) filtering algorithm was used to smooth the interpolated timeseries and suppress noise effects [44]. Finally, we obtained a cloud-free and gap-filled 10-day REP timeseries.

Sentinel-1 consists of two polar-orbiting satellites performing C-band synthetic aperture radar imaging. These satellites can provide dual-polarization observations, allowing them to acquire images independent of changing weather and environmental conditions. In this study, we used all available VV and VH polarization schemes of the C-band SAR Ground-Range-Detected (Sentinel-1 SAR GRD) data in IW mode from 2016, as provided by the 'COPERNICUS/S1_GRD_FLOAT' image collection of GEE. All Sentinel-1 images were processed to an angular-based radiometric slope correction using a Shuttle Radar Topography Mission digital elevation model (SRTM DEM) to reduce the impacts of local terrain on backscatter. The images were filtered using a refined Lee filter with a $7 \times 7$ moving window to mitigate the effects of "salt-and-pepper" noise. In addition, we added the VV/VH ratio in the analysis since previous studies have shown favorable results when applying this ratio in classifications of deciduous and broadleaf tree species [25]. Three radar indices were also computed: the modified radar vegetation index (MRVI) [45], the dual-polarization SAR vegetation index (DPSVIm) [46], and the normalized difference of the bands (NDI) [47].

Climatic conditions and environmental factors largely influence plant growth and tree species composition. The ecological processes of climate change affect the succession of dominant tree species and their distribution patterns. To understand the importance of environmental factors in our tree species classification, we utilized auxiliary variables describing topography, land surface temperature, and precipitation conditions. The MOD11A2 V6.1 product provides average 8-day land surface temperatures (LST) at a 1000-m resolution [48]. We composited this product into monthly average values. In addition, monthly average precipitation data at a 1-km resolution by OpenLandMap Precipitation Monthly were also used [49]. Previous studies have shown that adding topographic factors can effectively improve tree species classification results in mountainous areas [50]. The topographic data used herein were derived from an SRTM DEM at a resolution of 30 m; we extracted elevation, slope, and aspect parameters from these data.

Five types of features were extracted from multiple sensors to discriminate the tree species, which are listed in Table 1.

**Table 1.** Features used in this study.

| Feature Number | Acronym | Description |
| --- | --- | --- |
| Spectral (26) | SP | 10 basic Sentinel-2 bands (blue, green, red, red edge 4, NIR, SWIR 2). 16 vegetation indices |
| Texture (6) | TX | The red edge was selected to calculate 6 GLCM metrics: the sum average, correlation, dissimilarity, variance, contrast, and cluster shade. |
| REP_Time Series (37) | REP_TM | A 10-day time-series of the REP |
| SAR (6) | S1 | VV, VH, CR, and 3 radar vegetation indices: NDI, mRVI, and DPSVIm. |
| Environmental Factors (27) | Env | Topographic (elevation, slope, and aspect) Monthly mean precipitation Monthly mean land surface temperature |

2.2.2. Reference Samples

Forest species ground survey data collected in the Forest Management Inventory (FMI) provide information regarding the species compositions of forests. Each survey contains a ground survey and a forest stand survey. In the ground survey, the locations of patches are investigated, and the land use types, topography, soils, and other factors are determined for every patch. Each forest stand survey includes an analysis of the stand origin, species composition, stand age, stand stock, and other factors.

To obtain representative and accurate sample data, we proposed a filter flow based on the original survey data for 2016. First, we obtained survey information about the study area by cropping small groups according to the boundaries of the study area. Second, forests and non-forests were masked, and subregions with pure species compositions were selected from the forest regions. A pure forest subclass was defined as a single vegetation class, homogeneous tree stands, dominant plant species, or land classes covering more than 65% of the area. Third, the standard deviation and mean values of the blue, NIR, SWIR, NDVI, and GREENNESS bands were calculated based on the median composite Sentinel-2 images of the 2016 growing season to set appropriate thresholds and exclude small classes that did not meet the required conditions. Finally, sample points for each tree species were generated from the purified polygons.

Seventy percent of the ground-sample points were used to train the classification model, and the remaining 30% were used to validate the model (Figure 2). The following nine tree species were identified in our study area for further analysis: *Pinus kesiya var. langbianensis* (PKV), other broadleaf species (OB), *Hevea brasiliensis* (HB), *Eucalyptus robusta* (EL), *Pinus yunnanensis* (PY), *Quercus* L. (QL), *Cunninghamia lanceolata* (Lamb.) Hook. (CL), *Betula* (BL), and *Alnus cremastogyne* Burk. (ACB).

*2.3. Design Data Cases and Feature Selection*

Seven data cases were designed to assess the performance of different data combinations. The first three individual data cases shown in Table 2 were designed to evaluate the importance of tree species classification. The rest four data combination cases were designed to help understand the synergism of optical, SAR and environmental data on tree species classification.

Feature selection is a nonnegligible step in machine learning classification tasks when dealing with high-dimensional features. However, features from multi-modal data are not essential to improve the classification accuracy due to the existence of the "dimension disaster" [51]. Many methods have been proposed to select feature subsets, such as recursive feature elimination (RFE), Boruta, and the Gini Index [52,53]. However, most methods of feature selection do not consider the co-linearity among variables. Therefore, the relevance hierarchical clustering method was used to retain the optimal features to minimize redundant information caused by the covariance between features and decrease the computation time [11]. This hybrid method helps to identify redundant features by

clustering. First, we assessed the feature importance of all features according to the Mean Decrease Impurity index (MDI) produced by the RF classifier. The MDI measures the decrease in the Gini purity of each feature for each decision tree to determine the importance of the feature. Second, hierarchical clustering of a certain number of features on the Spearman rank-order correlations was conducted. The features with high correlation were grouped into clusters. The dendrogram produced by hierarchical clustering generates an immediate level organization of the feature space. Finally, each cluster's variables with the highest feature importance scores were retained. For cases 2,4,6, and 7, 32 out of the 69 variables, 31 out of the 75 variables, 32 out of the 96 variables and 32 out of the 102 variables were retained by feature selection, respectively. Considering that number of features from Sentinel-1 and environmental data is small, we did not perform feature selection on these datasets used alone.

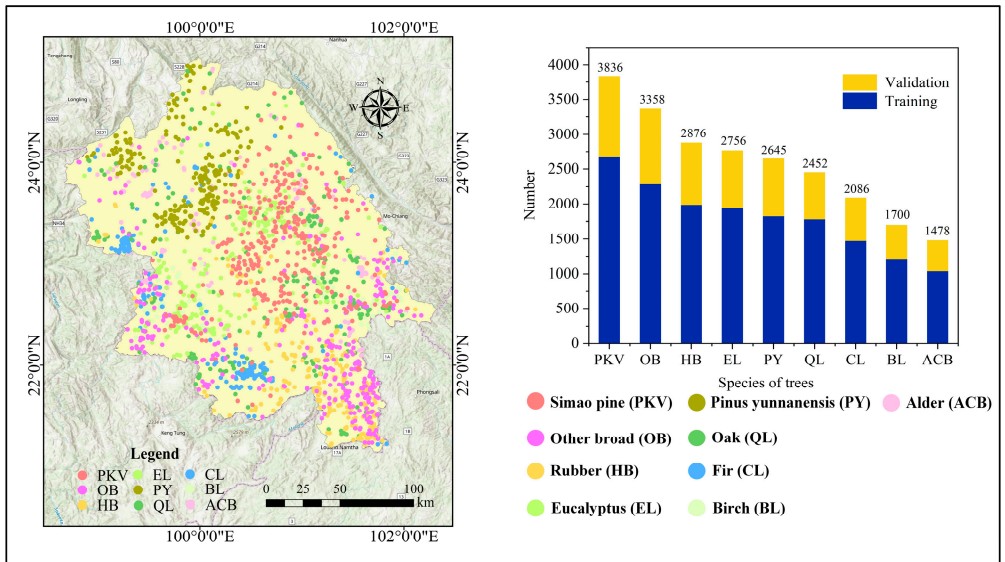

**Figure 2.** Spatial distribution and counts of field reference data points.

**Table 2.** Different data cases are designed for tree species classification.

| Serial Number | Abbreviation of Data Cases | Data Source | Number of Features |
|---|---|---|---|
| Cases 1 | S1 | Sentinel-1 | 6 |
| Cases 2 | S2$_{(SP+TX+REP\_TM)}$ | Sentinel-2 | 69 |
| Cases 3 | Env | Topographic, temperature, precipitation | 27 |
| Cases 4 | S2$_{(SP+TX+REP\_TM)}$ + S1 | Sentinel-1, Sentinel-2 | 75 |
| Cases 5 | S1 + Env | Sentinel-1, topographic, temperature, precipitation | 33 |
| Cases 6 | S2$_{(SP+TX+REP\_TM)}$ + Env | Sentinel-2, topographic, temperature, precipitation | 96 |
| Cases 7 | S2$_{(SP+TX+REP\_TM)}$ + S1 + Env | Sentinel-1, Sentinel-2, topographic, temperature, precipitation | 102 |

Note: See Table 1 for the meanings of abbreviations.

*2.4. Classification*

2.4.1. Classification Model and Assessment

In this study, we selected three nonparametric machine learning models, including RF, SVM, and extreme gradient boosting tree (XGBoost), for tree species classification. These algorithms have been widely adopted due to their reliable performance and stability in various remote sensing applications [9,15].

The RF algorithm is a machine learning method combining bagging ensemble learning theory with a random subspace approach [54]. The RF averages the prediction of each decision tree to obtain the final prediction. The RF is more robust and accurate than many conventional classifiers, such as the maximum likelihood, single decision trees, and single-layer neural network classifiers [37]. We adjusted one parameter in the RF within the GEE platform: the numberOfTrees: this parameter determines the number of binary classification and regression trees (CARTs) used to build the RF model. When the number of trees increases, the accuracy increases, and the computational cost increases linearly. We adjusted the numberOfTrees (the number of internally grown trees) and tried different settings of 100, 120, 150, and 180 according to some previous works and the number of features used in this study. Finally, we set the number of trees to 150. The other three parameters, including the number of variables per split, the fraction of input to bag per tree, whether the classifier should run in out-of-bag mode and random seed parameters, were set by default in GEE.

The SVM is a machine learning algorithm built on Vapnik-Chervonenkis dimensional (VC) statistical learning theory and the structural risk minimization criterion [55]. SVM is widely used in remote sensing image classification tasks and has a strong generalization capability and robustness in solving nonlinearity, small-sample classification, and high-dimensional data problems. The radial basis function (RBF) was chosen as the kernel function for the SVM in this study, and two other parameters, gamma, and cost, were manually adjusted several times. Here, we set the gamma parameter to 0.5 and the cost parameter to 8.

The XGBoost algorithm is similar to the gradient boosting framework; this model combines a linear model and a tree learning algorithm to form a new efficient boosting algorithm. It is optimized and improved on the base algorithm and has demonstrated excellent performance in remote sensing applications [56]. In this study, we adjusted and set the numberOfTrees to 45, the shrinkage to 0.006, and samplingRate to 0.7.

A confusion matrix is often used to evaluate the accuracy of classification task results and can visualize classification and omission errors in each category. In addition, various accuracy measures can be calculated from a confusion matrix. The classification results obtained herein were evaluated by building a confusion matrix for the analyzed tree species and by calculating the overall accuracy, Kappa, user's accuracy (UA), and producer's accuracy (PA) as evaluation metrics based on ground validation samples.

2.4.2. Overview of the Proposed Method

The methodology and experimental setup of this study are demonstrated in Figure 3. The framework consists of the following steps. (1) Sample data filtering; (2) Data pre-processing; (3) Feature extraction; (4) Classification case design and feature reduction; (5) Accuracy evaluation and mapping.

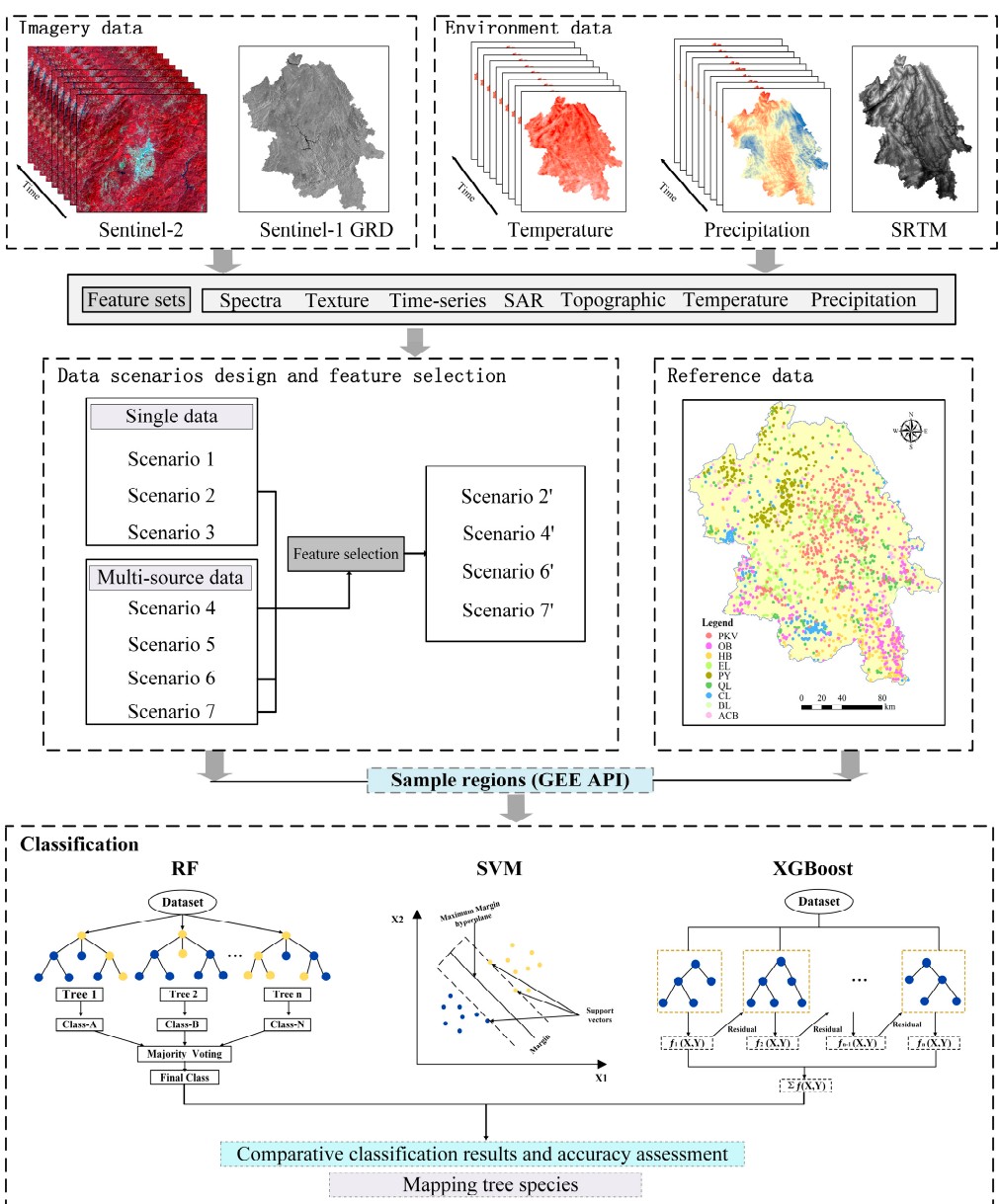

**Figure 3.** Flowchart of the proposed method.

## 3. Results

### 3.1. Classification Results

The performances of the 11 different cases in the three machine learning models are shown in Figure 4. The integration of spectral, texture, time-series, SAR, and environmental factors and the RF classifier provided the best classification results with an overall accuracy of 77.98% and Kappa coefficient of 0.75, followed by the $S2_{(SP+TX+REP\_TM)}$ + Env combination (the combination of Sentinel-2-derived features and environmental factors), with an overall accuracy of 76.29–77.23% and Kappa coefficient of 0.73–0.74. The poorest performance was obtained using only Sentinel-1 data on RF, with OA 27.49% and Kappa 0.16. The $S2_{(SP+TX+REP\_TM)}$ + S1 combination slightly improved compared to $S2_{(SP+TX+REP\_TM)}$. Compared to the features of Sentinel-2, combining environmental factors resulted in substantial improvement, with an overall accuracy increase of 7.19–13.78%. There was no significant difference in accuracies among using feature sets after feature selection compared to all features. Feature selection significantly reduced the cost of model training time while

maintaining similar accuracy. The feature sets of the four cases after feature selection are presented in Table A6.

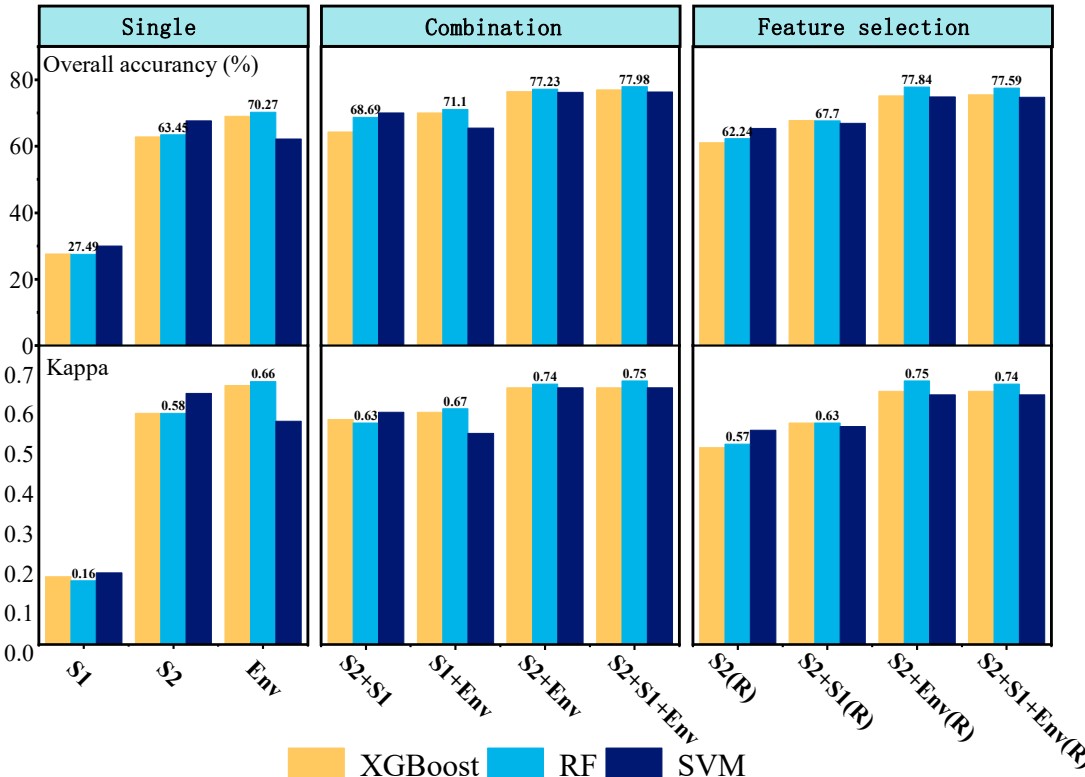

**Figure 4.** Classification accuracies were obtained from different classification cases and three classifiers.

Considering only features from Sentinel-2 did not yield satisfactory results when classifying tree species in large mountainous areas. To further understand the impacts of Sentinel-2 in mapping specific tree species, we additionally computed the confusion matrix for the individual and combination of three types of features. Table A1 shows the overall accuracy (OA) of 64.18% and Kappa coefficient of 0.59 achieved using only spectral features in the SVM. Table A2 shows the confusion of individual species obtained by adding texture features to the spectral features ($S2_{(SP+TX)}$). The texture features significantly improved only the birch and fir classification results, while the confusion between some other species was aggravated instead of improved. However, texture information usually does not effectively improve the separability of all classes [35]. Although previous studies have shown that time-series features can improve tree species classification, these features did not result in the expected results in this research. As shown in Table A3, for each species, the combination of spectral, texture, and time-series features improved the classification accuracy compared to the combination of only spectral and texture features, especially for Yunnan pines and birch. Time-series features improved the PA and UA of Yunnan pines by 9.02% and 3.16%, respectively and the PA and UA of birch by 7.42% and 14.42%, respectively.

The results obtained using Sentinel-1 data alone were not informative; using only Sentinel-1 data yielded the lowest OA at 27.49%. However, adding Sentinel-1 data based on Sentinel-2 (case 4) was improved slightly, with a resulting OA of 69.99% and a Kappa coefficient of 0.66 in the SVM model (Table A4). Specifically, with the integration of SAR variables, the impact on the Yunnan pine results was more significant, with the PA and UA values increasing by 10.72% and 4.71%, respectively. The OA obtained under case 6 (S1 + Env) also showed low improvement compared to that under case 3 (Env), as presented in Figure 4.

Environmental features allowed the optimal accuracy to be obtained with the use of a single data source, with an overall accuracy approximately 6% higher than that obtained for case 2 (Sentinel-2). Case 5 and Case 6 obtained overall accuracies of 71.10% and 77.23%, respectively, with the RF classifier model, respectively. Environmental features increased the PA of Yunnan pine by 14.86% and UA of Yunnan pine by 15.38% compared to the S2$_{(SP+TX+REP\_TM)}$ + S1 combination (Table A5). Misclassifications among broadleaf species are the most frequent, especially among alder, birch, oak, and other broadleaf species. However, the needleleaf and broadleaf species were effectively distinguished from each other; e.g., the two major broadleaf classes, rubber, and other broadleaf species were confused with Yunnan and Simao pines to a much lesser extent (Table A5).

### 3.2. Mapping of the Tree Species Classification Results

We chose the best-performing RF classification derived from the feature set combining all data sources to visualize the tree species map (Figure 5). Simao pine is mainly distributed in the eastern and northeastern regions of the study area. Moreover, the southern area is dominated by rubber and other broadleaf trees; these results are the situation. The Xishuangbanna region (the southern region of the study area) has a tropical and subtropical monsoon climate with sufficient sunshine and abundant rainfall. It is an important rubber-producing region in China. Yunnan pine is mainly concentrated in the northwestern region and is mixed with a portion of alder. Eucalyptus is relatively concentrated in the Puer city area (in the central part of the study area), and the area of fir makes up the largest proportion of all tree species, covering almost the entire study area. Oak species are widely distributed but relatively scattered across the study area. More particularly, the spatial location of birch species has a precise latitudinal distribution along the east-west continuum spanning the study area.

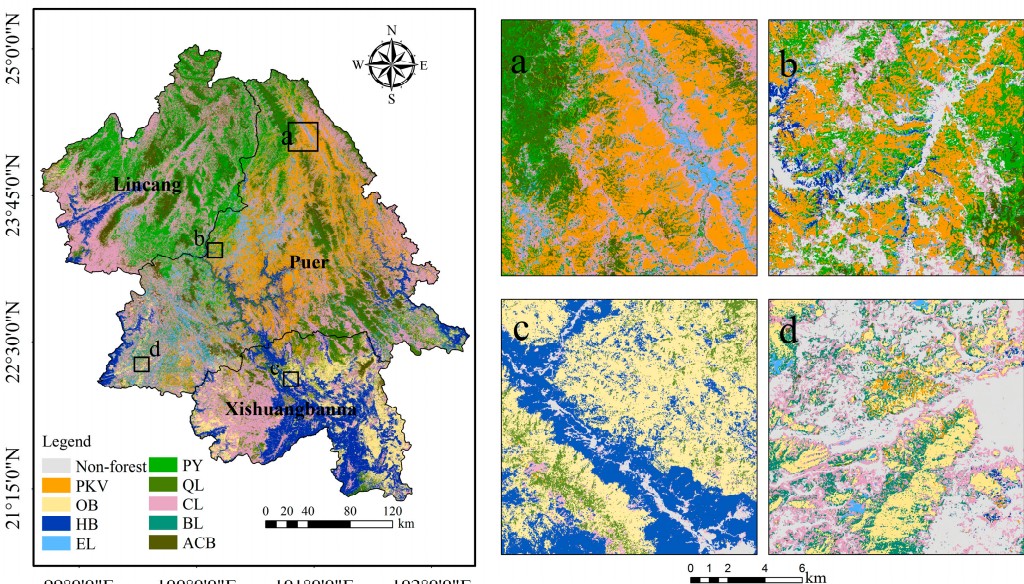

**Figure 5.** Mapping tree species with all feature combinations. (**a–d**) show the spatial details of the tree species map.

In this study, we produced tree species maps in a large subtropical and tropical mountainous area using three machine learning classifiers on a cloud-computation platform. Seven main data cases were designed to evaluate the performance of single and different data (Sentinel-1, Sentinel-2, and environmental data) combinations. The best accuracy (77.98% OA) of mapping tree species was achieved by using all data sources on RF algorithms. We obtained similar classification accuracies with previous studies for tree species mapping and had a larger study area above 70,000 km$^2$ than these study areas [25,57,58].

Figures 6 and 7 describe the partial results obtained under three classification cases in the selected area compared to the markers obtained from the ground survey. We selected two subregions of the classification results from seven cases for comparison with the ground data. Figure 6i shows the map almost completely obtained for the study region; the map almost agrees with the ground survey results. However, the area illustrated in Figure 6 only shows individual tree species with high classification accuracy, and Figure 7 shows the mapping of oak species that cannot be provided as a reference. The area in Figure 7 is dominated by oak and Simao pines. The misclassification between the two species is significant, with all the oak being mistaken for Simao pines in the cases with only environmental data. One possible reason is the coarse spatial resolution (1 km) of climate data; these datasets could not provide more detailed spatial information thanSentinel-2 data in complex vegetation cover areas. Better visual results are obtained with the Sentinel-2 data.

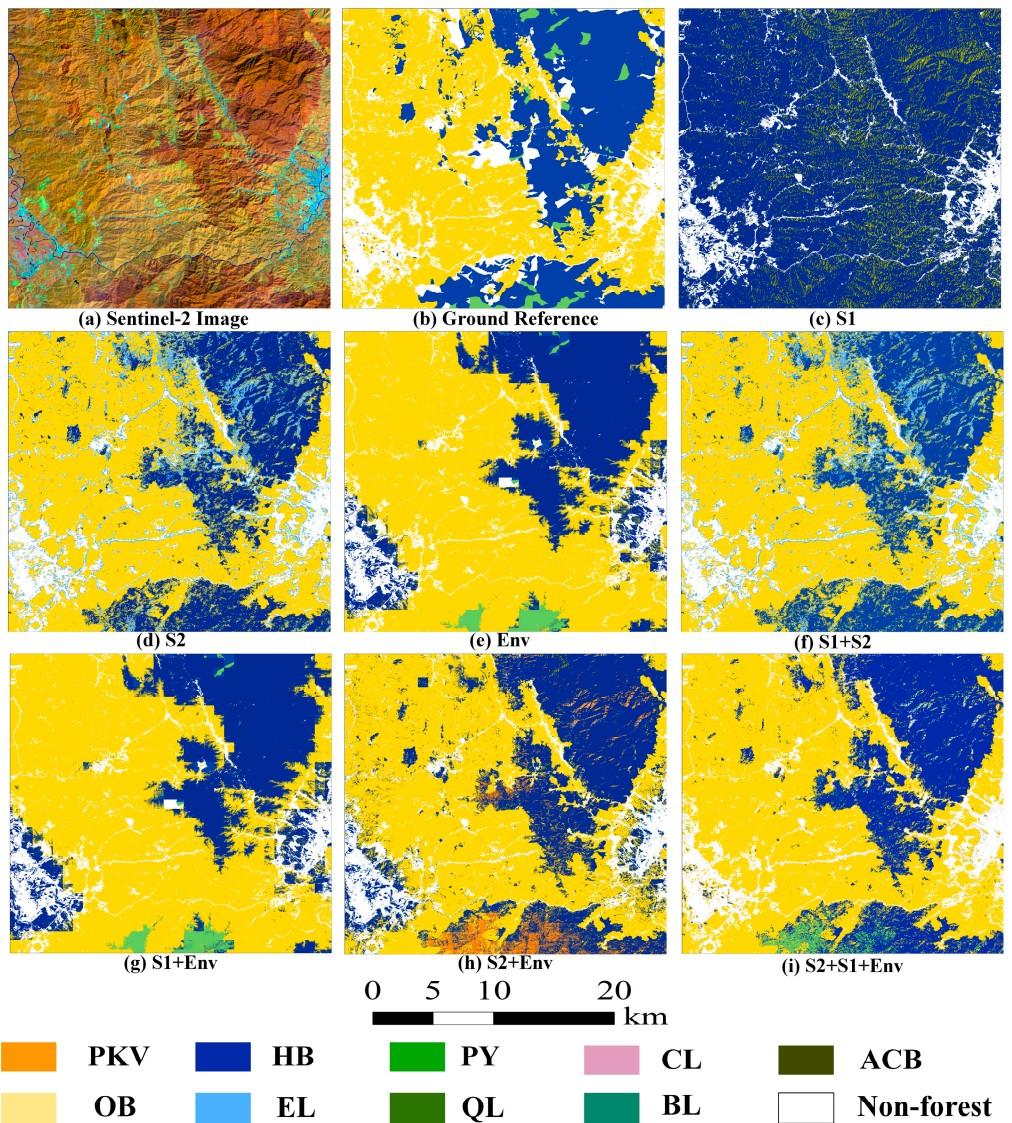

**Figure 6.** Detailed view of the tree-species maps with better interpretation derived under different cases. (**a**) false-color image synthesized from the SWIR1, NIR, and blue bands; (**b**) ground survey labels; and (**c**–**i**) RF maps predicted based on the 7 feature sets.

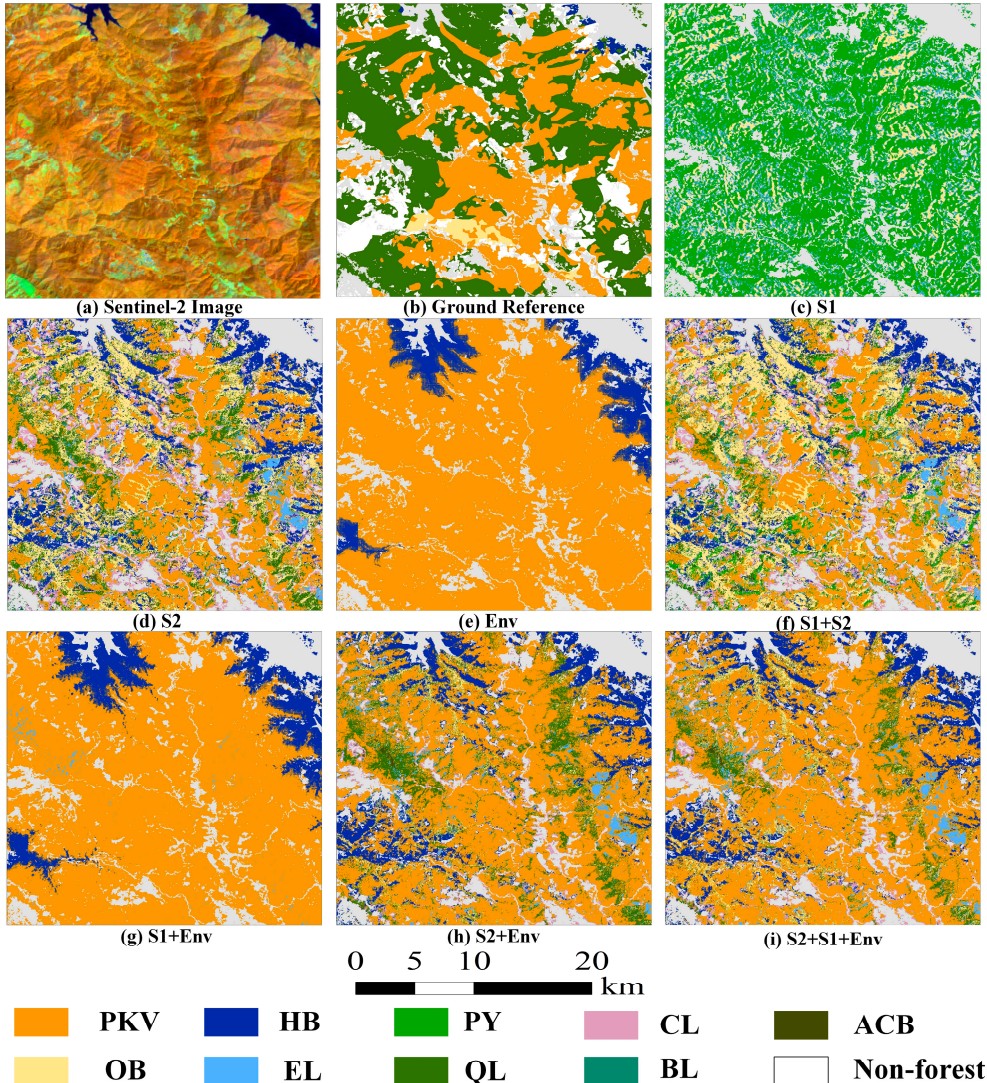

**Figure 7.** Detailed view of the tree-species maps with worse interpretation derived under different cases. (**a**) false-color image synthesized from the SWIR1, NIR, and blue bands; (**b**) ground survey labels; and (**c–i**) RF maps predicted based on the 7 feature sets.

In this study, natural forests cover the entire study area and are extensively distributed, while planted forests are relatively concentrated. Significant differences exist in spatial distribution and species composition between natural and plantation forests (Figure 8). However, plantation forests are dominated by Simao pine and rubber, with high accuracies of over 90%. For example, rubber is mainly distributed in the Xishuangbanna region and mainly planted manually, with a relatively concentrated distribution and containing more pure pixels. Although the multi-modal data improved the classification of each tree species compared to the single data, some classes are still assigned at a relatively low accuracy, such as oak, with an OA of only 53.92%. Two reasons may explain this low accuracy. The first is that approximately 97% of the oak species are from natural forests, which are fragmented throughout the study area, aggravating the mixed pixel situation. The second is that the genus oak contains many species with large morphological differences. The tree species classification results only depend not only on their physiological characteristics but also on their origin and later anthropogenic disturbances. These findings may provide useful information for future tree species mapping.

### 3.3. Feature Importance Assessment

To understand the separability among tree species in different classification cases, the Jeffires–Matusita (JM) distance method is considered suitable for expressing category separability [59]. The JM results are classified into four ranks, with values from the 1.9–2.0 range representing strong separation, values from 1.8–1.9 representing good separation, values from 1.7–1.8 representing weak separation, and values from 0–1.7 representing poor separation, as illustrated in Figure 9. For each tree species, 1000 sample points were randomly selected to calculate the JM distance. In this figure, panels a-f represent the calculated results of cases 1–7. From panel a, Sentinel-2 data made only Yunnan Pine somewhat separable from the other species. With the integration of SAR features, the separability of some tree species increased. However, it was still difficult to distinguish most broadleaf species, such as rubber and oak trees, from one another. The best results were acquired from case 7 with a combination of all data sources.

Considering the variability in the tree species classification results across feature sets, we also used a feature importance assessment to determine which features contributed most to nine tree species classifications. Feature importance scores were calculated internally by the RF algorithm on the GEE platform. The Gini importance score describes the relative importance of features in the classification process. Here, we ranked the importance of 102 features from case 7 (the best features combination), as shown in Figure 10. The topmost feature was the NDTI, followed by elevation. Among the first 30 features, there were only 2 spectral bands and 6 indices; the remaining features were environmental factors. Environmental features played an essential role in the classification of tree species in this study, among which the monthly mean precipitation and monthly mean land surface temperature accounted for significant proportions. Furthermore, the top-ranking temperature and precipitation factors were mainly distributed in May, June, July, and August, which interestingly span the rainy season in the study area.

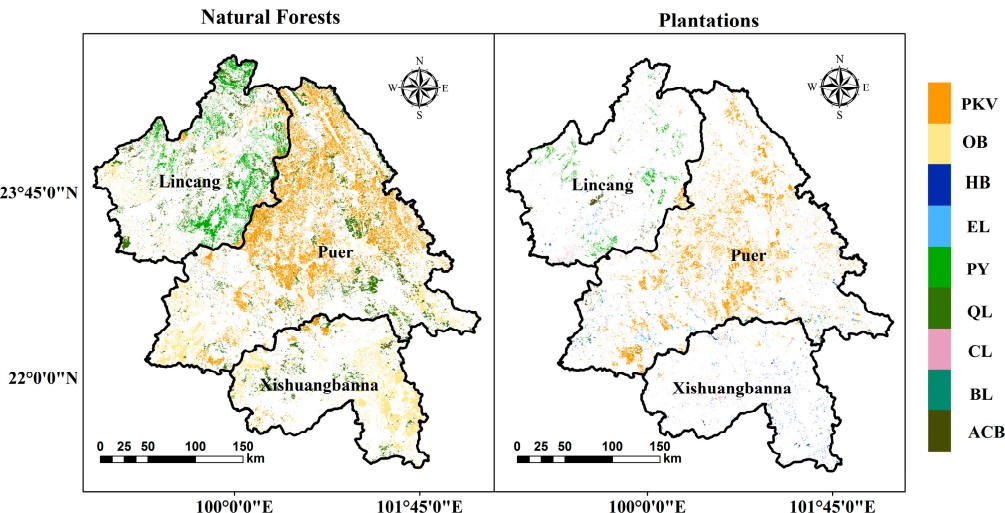

**Figure 8.** Spatial distribution of nine tree species in natural and plantation forests.

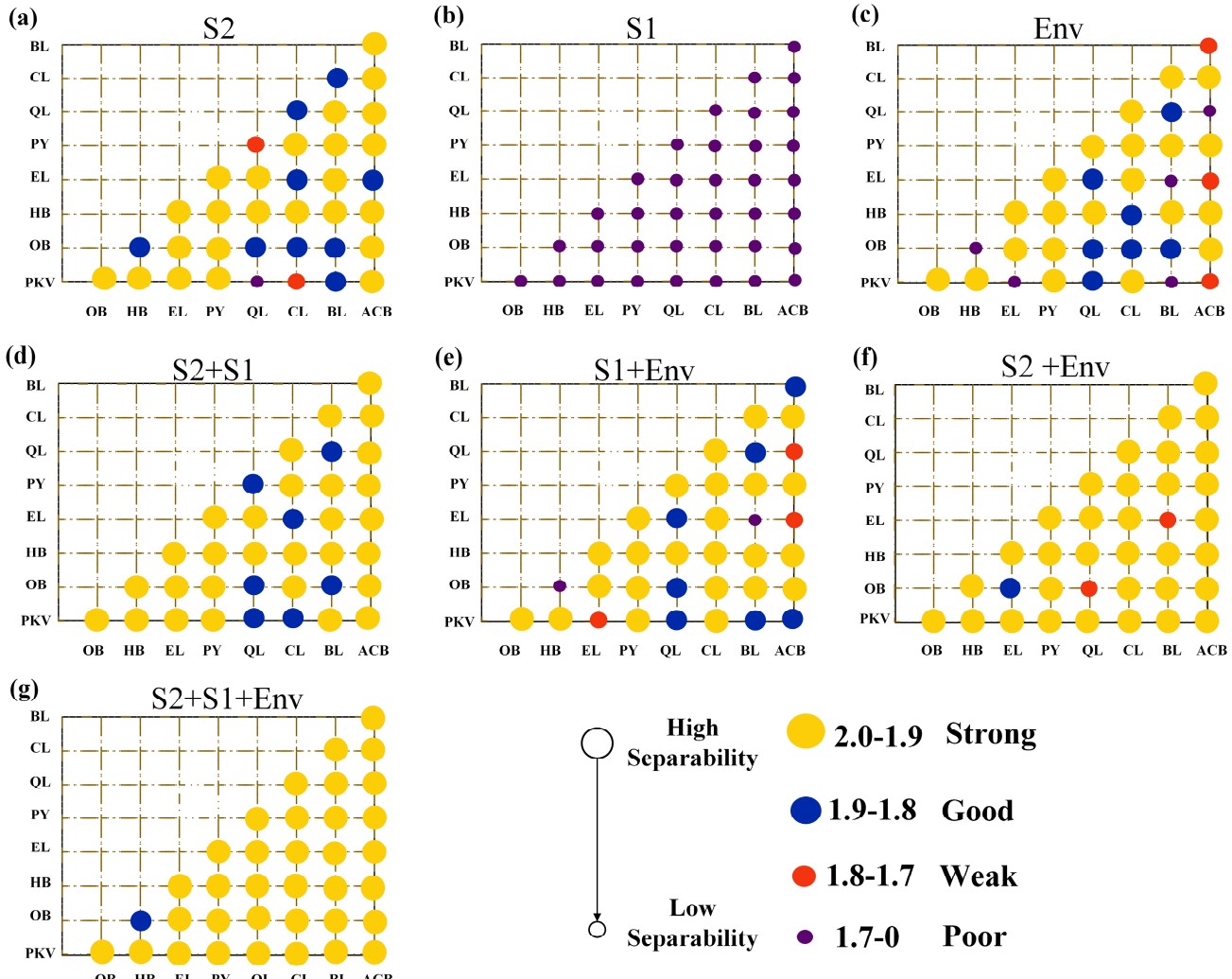

**Figure 9.** Class separability results derived based on the JM distance: (**a**–**g**) represent case 1–case 7.

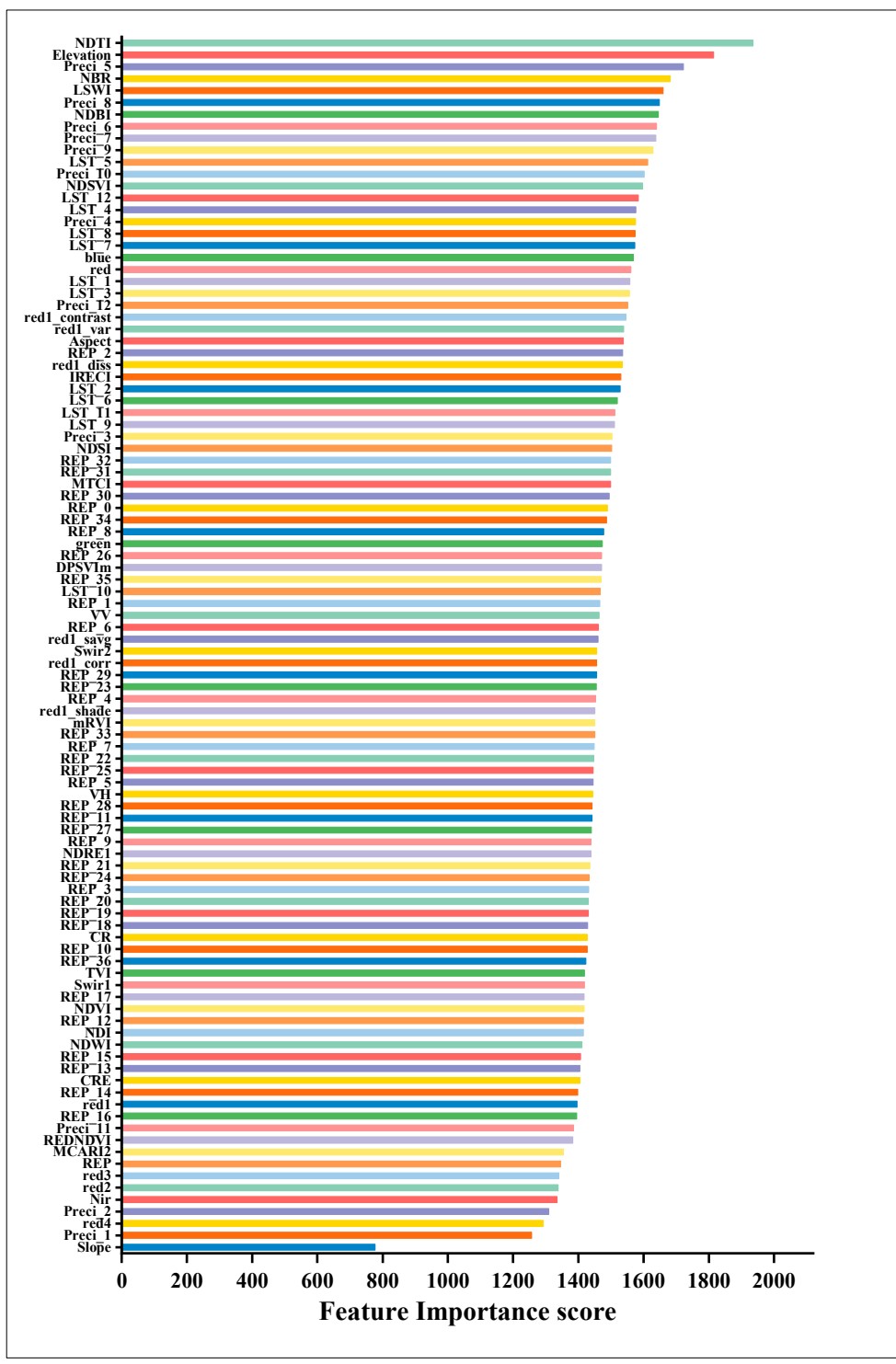

**Figure 10.** RF feature importance results based on case 7 for tree species classification. The feature abbreviations in the figure can be found in Section 2.2.1. With LST as a prefix representing land surface temperature and with "Preci" as a prefix representing precipitation, the number after them represents the month. The number after REP represents the time information in steps of 10 days.

## 4. Discussion

In this study, we evaluated the utility of multi-modal data from different remote-sensing sensors, such as optical, SAR and environmental datasets. The Sentinel-2 data used in the study produced spectral, texture and time-series features. We calculated 16 vegetation indices that have been confirmed in previous studies to be related to vegetation growth, senescence, and vegetation water content but did not discuss whether they are useful for discriminating between tree species. As shown in A1, using spectral features alone cannot produce sufficiently high classification accuracies for all tree species. The relatively narrow and few spectral bands limit their ability to distinguish tree species, especially in subtropical and tropical regions with abundant vegetation types. Nevertheless, Sentinel-2 worked relatively well for certain classes with significant spectral differences, such as rubber and Simao pines. As shown in A2, adding texture features did not contribute to an improved classification ability, possibly due to spatial resolution limitations. Most studies on remote sensing classification that used high spatial resolution images stated that texture features had outstanding performance [60,61]. In addition, the bands in which the texture features were calculated and the window size affected the classification accuracy of the tree species. The combination of texture features applicable for classifying tree species varies among tree species. This implies that it is difficult to find a universal texture feature combination applicable to all tree species. However, the usefulness of texture features is related to the size of the selected window, which depends on the spatial resolution of the imagery. The relatively short revisit period and high spatial resolution of these data can facilitate the extraction of spectral features with temporal information for each type of tree species. Significant effects were observed for two coniferous species, Yunnan and Simao pines, and a slight performance increase was observed for several broadleaf species (Table A3). Yunnan and Simao pines are mutually misclassified. The probable reason is that both species are evergreen trees of the genus Pinus and have remarkable similarities concerning their physiological structures and morphology, leading to difficulties distinguishing them in terms of spectral and textural information. Meanwhile, tree species' phenological differences are not obvious, and it is difficult to distinguish them in this manner, especially among broadleaf species. Although we performed a median composite of the available observations to mitigate the influence of no-data pixels after cloud masking, the true surface reflectance of such composited pixels is still inaccurate.

Sentinel-1 sensors can observe the earth under all-weather conditions, and they are often used as a complement to Sentinel-2. The backscattering features and radar vegetation indices obtained from Sentinel-1 data exerted a 4% improvement in the tree species classification results. The C-band generally provides information from a combination of ground backscatter after canopy attenuation and backscatter information directly from the canopy. Specifically, the VH polarization mode is dominated by the ground surface, but complex volume scattering also occurs in the canopy [62]. This situation occurs mainly in forested areas with low vegetation densities. It is challenging to differentiate tree species using Sentinel-1 data alone in regions with dense tree canopies and complex forest structures. Although many studies have shown that radar indices effectively distinguish vegetation and crops, the importance levels of the features analyzed herein indicate that these indices are not dominant in classifying the nine tree species. Although the contribution of Sentinel-1 was relatively low for tree species classification, it was still helpful in improving the accuracy of the classification of Simao pine and Yunnan pine. This is possibly related to the difference in canopy surface roughness between conifers and deciduous trees (the canopy surface of deciduous is smoother than that of conifers) [25]. However, combining Sentinel-1 and Sentinel-2 data did not reduce the omission error of oak, indicating that the integration of these data could not provide sufficient information to accurately distinguish all species.

The feature importance analyzed in this study demonstrated the significance of topographic features; elevation ranked second among all features. A box-and-whisker plot was created to show the elevational differences among tree species according to each type of tree (Figure 11). The results reflect that oak and other broadleaf species have the

most extensive elevational range, growing between 500 and 3100 m. These two classes also contain many other species, such as the experimental combination of hemp oak and green oak, considered an oak species. The rubber elevation distribution ranges from 500–1300 m, the smallest elevation range among the analyzed classes, with an average elevation of 900 m. Tree species' distributions strongly correlate with elevation, and the environmental conditions in different elevational ranges vary [63].

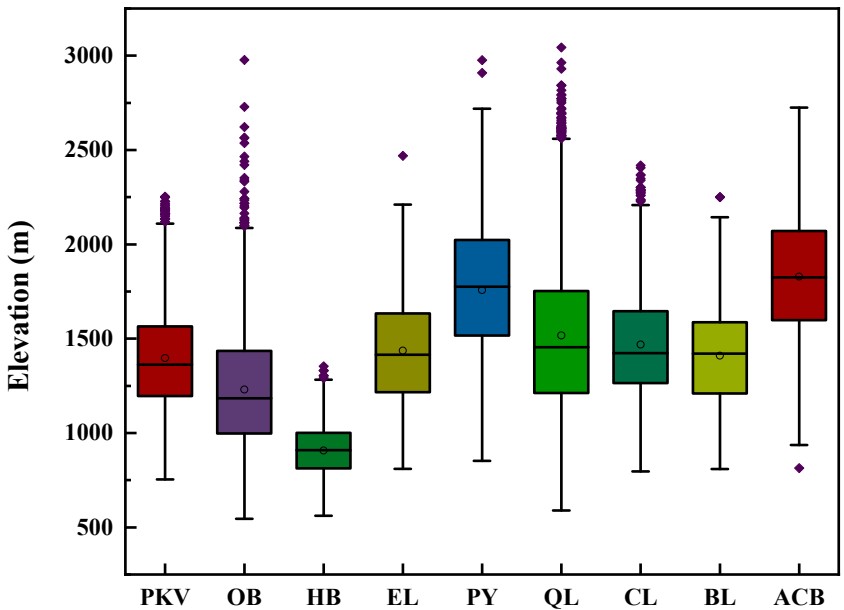

**Figure 11.** Box−and−whisker plot of tree species elevation distributions.

Tree species distributions are usually subject to environmental conditions, such as abiotic conditions, including soil, temperature, and precipitation [64]. Climate and remotely sensed data are mutually complementary in tree species classification tasks. In addition to the contribution of topographic factors, the precipitation and land surface temperature factors also played essential roles in the classification. To further understand the relationships between land surface temperature and tree species and between precipitation and tree species, we calculated the average monthly land surface temperature and precipitation conditions for each tree species in the study area, as shown in Figures 12 and 13. The land surface temperature statistics for the two conifer classes of Simao pine and Yunnan pine are lower than those for the broadleaf species. Both species are usually distributed on mountains at relatively high elevations. In addition, alder shows lower acclimatization-temperature conditions, consistent with the fact that alder is suitable for growing at an average annual land surface temperature of 15~18 °C. The monthly land surface temperature averages of rubber than the other tree species, which is related to the growing environments of alder, which are mainly located in areas near the equator. Toledo et al. noted a significant response of temperature when determining the flora to which a species belongs in their study of flora composition and its relationships with environmental factors [65]. In addition, the annual precipitation and growing season precipitation in coniferous forest areas (Simao pine and Yunnan pine) are less than those in broadleaf forests. The spatial distribution of tree species in this study area reflects the spatial heterogeneities in topography and climate conditions associated with species composition.

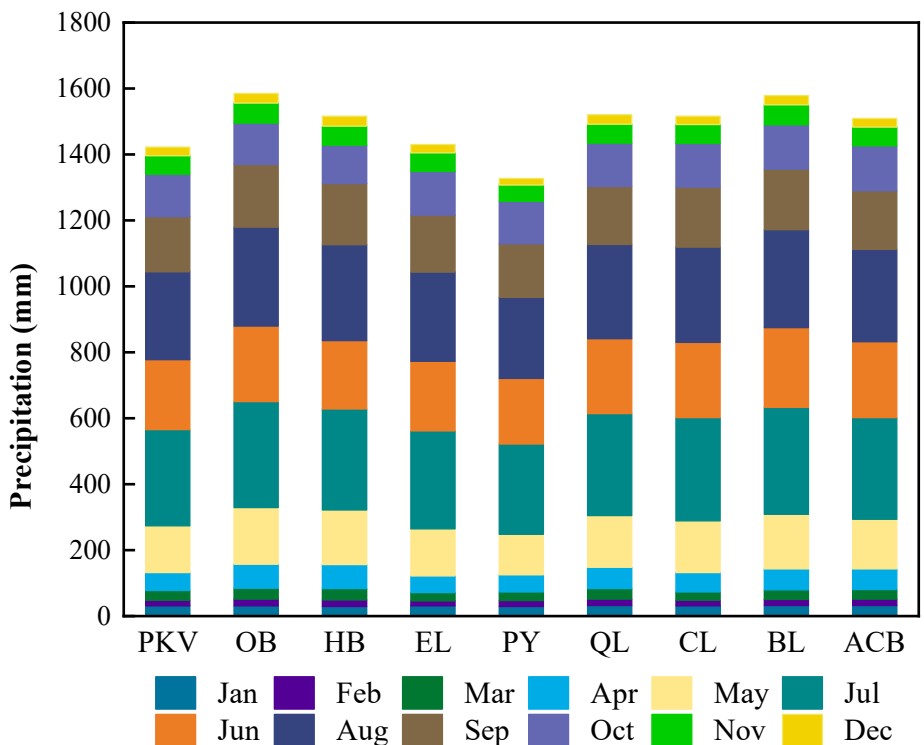

**Figure 12.** Average monthly precipitation statistics of different tree species in the study area.

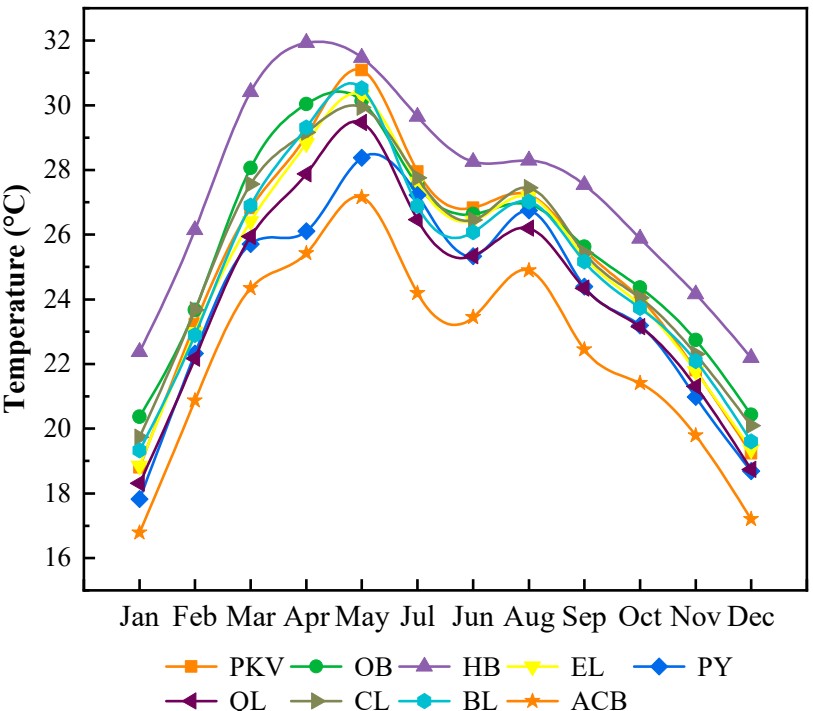

**Figure 13.** Average monthly land surface temperature statistics of different tree species in the study area.

The best accuracy of 77.98% was obtained by combining Sentinel-1 and Sentinel-2 imagery with environmental factors in the RF. Among the different machine learning algorithms, we found that the RF classifier performed slightly better than the SVM and XGBoost classifiers. The optimal performance of the RF has also been reported in past tree species classification studies [66]. In this study, the performance differences of the

analyzed classifiers were not sufficiently noticeable, thus indicating the stability of the selected features. In addition, the quality and number of samples per class and the tuning of hyperparameters both impacted the models' performance. Thus, the three classifiers used in our study are well-suited for tree species classification tasks. In addition, too high feature dimensions increase the training time. Therefore, determining how to filter the important features and maintain high accuracy is essential for producing large-scale maps. Moreover, the selection of multi-modal data should consider the cover type and composition in the study area.

## 5. Conclusions

In this study, we designed different classification cases based on freely available data to explore their potential use for mapping tropical and subtropical mountainous tree species classification. Synergistic multi-modal data represent a new opportunity to improve mountainous tree species classification. The results showed that Sentinel-2 data had a nonnegligible effect on the tree species classification results. SAR data could compensate for the lack of Sentinel-2 images, but it was less capable of distinguishing all tree species. Environmental variables are significant for tree species classification, which has rarely been explored in previous studies, especially in a mountainous area. Furthermore, topographic features exhibited an advantage in classifying tree species in mountainous areas, especially elevation features. The environmental conditions of different species vary significantly. Although with a lower spatial resolution, precipitation and land surface temperature data play considerable roles in the mapping of tree species, especially in distinguishing coniferous species from broadleaf species. In conclusion, considering a combination of multi-modal data sources provided accurate information regarding the distributions of tree species, especially when considering factors related to the growing environments of tree species. The final tree species map was obtained from synergistic of all data and achieved an overall accuracy of 77.98%, demonstrating that synergistic multi-modal data has the potential to map tree species in complex mountain vegetation coverages and help the forestry department in updating the forest information. Both the remote sensing and environmental data used in this experiment are freely available on the GEE platform, and the production of the products is convenient, thus allowing the transfer of the classification strategy proposed herein to larger study areas. Mapping a larger range of tree species, acquiring an accurate sample, and selecting optimized features are necessary. However, mapping and understanding the detailed spatial distributions of tree species requires further exploration by combining other data sources; these tasks are essential for developing future forest management strategies.

**Author Contributions:** Conceptualization, L.W.; methodology, P.Z. and P.F.; software, F.D. and P.Z.; validation, G.O. and W.X.; formal analysis, P.Z.; data curation, G.O.; writing—original draft preparation, P.Z.; writing—review and editing, Q.D. and L.W.; visualization, P.Z.; supervision, Q.D.; project administration, L.W.; funding acquisition, L.W., Q.D. and W.X. All authors have read and agreed to the published version of the manuscript.

**Funding:** This research was funded by the National Natural Science Foundation of China (Under Grants. 32160369, 31860182, 41961053 and 32060320), partly supported by the Key Development and Promotion Project of Yunnan Province under Grants 202202AD080010, Research Foundation for Basic Research of Yunnan Province (202101AT070039), Joint Special Project for Agriculture of Yunnan Province, China (202101BD070001-066), and "Ten Thousand Talents Program" Special Project for Young Top-notch Talents of Yunnan Province (YNWR-QNBJ-2020047).

**Data Availability Statement:** Not applicable.

**Acknowledgments:** The authors would like to thank the Google Earth Engine team for their wonderful work and service.

**Conflicts of Interest:** The authors declare no conflict of interest.

## Appendix A

Classifications' confusion matrix on RF is reported here to support the main results.

**Table A1.** Confusion matrix derived using only Sentinel-2 spectral features.

| | | Reference | | | | | | | | | |
|---|---|---|---|---|---|---|---|---|---|---|---|
| | | PKV | OB | HB | EL | PY | QL | CL | BL | ACB | UA (%) |
| S2(SP) classification | PKV | 819 | 50 | 20 | 77 | 146 | 61 | 114 | 52 | 17 | 60.71 |
| | OB | 61 | 738 | 78 | 35 | 14 | 258 | 31 | 98 | 36 | 53.82 |
| | HB | 5 | 70 | 736 | 7 | 7 | 28 | 17 | 14 | 4 | 82.05 |
| | EL | 13 | 9 | 1 | 631 | 10 | 5 | 10 | 22 | 2 | 90.40 |
| | PY | 122 | 14 | 1 | 18 | 516 | 55 | 16 | 9 | 49 | 62.69 |
| | QL | 78 | 97 | 2 | 14 | 41 | 188 | 16 | 36 | 42 | 38.47 |
| | CL | 22 | 14 | 17 | 10 | 16 | 16 | 339 | 18 | 13 | 72.71 |
| | BL | 20 | 46 | 0 | 24 | 10 | 31 | 18 | 228 | 17 | 57.07 |
| | ACB | 13 | 28 | 0 | 8 | 61 | 46 | 13 | 22 | 258 | 56.08 |
| | PA (%) | 71.03 | 69.23 | 86.08 | 76.57 | 62.85 | 27.32 | 55.48 | 45.69 | 58.90 | |
| | | OA: 64.18% | | | Kappa: 0.59 | | | | | | |

Note: UA: user's accuracy, PA: producer's accuracy, OA: overall accuracy; see Section 2.2 for the abbreviations of tree species names.

**Table A2.** Confusion matrix derived using Sentinel-2 spectral and texture features.

| | | Reference | | | | | | | | | |
|---|---|---|---|---|---|---|---|---|---|---|---|
| | | PKV | OB | HB | EL | PY | QL | CL | BL | ACB | UA (%) |
| S2(SP+TX) classification | PKV | 836 | 59 | 14 | 56 | 171 | 79 | 69 | 38 | 13 | 62.48 |
| | OB | 61 | 695 | 100 | 29 | 11 | 208 | 20 | 88 | 26 | 56.14 |
| | HB | 6 | 75 | 729 | 5 | 5 | 23 | 29 | 12 | 4 | 82.09 |
| | EL | 14 | 18 | 1 | 642 | 14 | 12 | 11 | 24 | 5 | 86.64 |
| | PY | 103 | 10 | 0 | 29 | 490 | 49 | 62 | 16 | 55 | 60.20 |
| | QL | 75 | 109 | 1 | 20 | 50 | 230 | 23 | 53 | 42 | 38.14 |
| | CL | 33 | 8 | 6 | 15 | 28 | 17 | 353 | 18 | 18 | 71.17 |
| | BL | 14 | 60 | 2 | 18 | 11 | 24 | 14 | 231 | 19 | 58.78 |
| | ACB | 11 | 32 | 2 | 10 | 41 | 46 | 30 | 19 | 253 | 56.98 |
| | PA (%) | 72.51 | 65.20 | 85.26 | 77.91 | 59.68 | 33.43 | 57.77 | 46.29 | 57.76 | |
| | | OA: 64.11% | | | Kappa: 0.59 | | | | | | |

Note: UA: user's accuracy, PA: producer's accuracy, OA: overall accuracy; see Section 2.2 for the abbreviations of tree species names.

**Table A3.** Confusion matrix derived using only Sentinel-2 spectral and texture features and REP time-series features.

| | | Reference | | | | | | | | | |
|---|---|---|---|---|---|---|---|---|---|---|---|
| | | PKV | OB | HB | EL | PY | QL | CL | BL | ACB | UA (%) |
| S2(SP+TX+REP_TM) classification | PKV | 824 | 49 | 14 | 65 | 121 | 7 | 63 | 38 | 14 | 65.19 |
| | OB | 63 | 732 | 64 | 30 | 14 | 245 | 35 | 97 | 32 | 57.50 |
| | HB | 4 | 60 | 764 | 2 | 6 | 15 | 10 | 6 | 2 | 87.72 |
| | EL | 17 | 17 | 1 | 658 | 9 | 8 | 3 | 15 | 3 | 88.56 |
| | PY | 113 | 21 | 1 | 20 | 564 | 45 | 52 | 10 | 56 | 63.66 |
| | QL | 78 | 115 | 3 | 18 | 36 | 223 | 13 | 23 | 46 | 41.97 |
| | CL | 27 | 14 | 5 | 8 | 20 | 14 | 400 | 20 | 15 | 73.49 |
| | BL | 11 | 28 | 2 | 14 | 4 | 14 | 13 | 275 | 18 | 75.33 |
| | ACB | 16 | 30 | 1 | 9 | 47 | 45 | 22 | 15 | 262 | 57.75 |
| | PA (%) | 71.47 | 68.67 | 89.36 | 79.85 | 68.70 | 35.32 | 61.70 | 57.52 | 58.68 | |
| | | OA: 67.66% | | | Kappa: 0.63 | | | | | | |

Note: UA: user's accuracy, PA: producer's accuracy, OA: overall accuracy; see Section 2.2 for the abbreviations of tree species names.

**Table A4.** Confusion matrix derived using Sentinel-2 and Sentinel-1 data.

| | | Reference | | | | | | | | | |
|---|---|---|---|---|---|---|---|---|---|---|---|
| | | **PKV** | **OB** | **HB** | **EL** | **PY** | **QL** | **CL** | **BL** | **ACB** | **UA (%)** |
| S2(SP+TX+REP_TM) + S1 classification | PKV | 870 | 46 | 12 | 72 | 60 | 86 | 65 | 32 | 15 | 69.16 |
| | OB | 57 | 751 | 61 | 31 | 22 | 204 | 31 | 81 | 33 | 59.09 |
| | HB | 4 | 59 | 761 | 1 | 6 | 17 | 13 | 5 | 2 | 87.67 |
| | EL | 16 | 11 | 1 | 659 | 6 | 13 | 8 | 14 | 6 | 89.78 |
| | PY | 83 | 26 | 2 | 14 | 644 | 49 | 56 | 13 | 55 | 68.37 |
| | QL | 74 | 99 | 9 | 16 | 18 | 259 | 22 | 27 | 41 | 45.84 |
| | CL | 27 | 13 | 8 | 10 | 20 | 11 | 387 | 24 | 22 | 74.14 |
| | BL | 12 | 30 | 1 | 12 | 3 | 12 | 10 | 288 | 16 | 75.00 |
| | ACB | 10 | 31 | 0 | 9 | 42 | 18 | 18 | 15 | 248 | 60.49 |
| PA (%) | | 78.40 | 71.29 | 88.54 | 79.73 | 79.42 | 33.72 | 65.41 | 54.11 | 57.53 | |
| | | OA: 69.99% | | | Kappa: 0.66 | | | | | | |

Note: UA: user's accuracy, PA: producer's accuracy, OA: overall accuracy; see Section 2.2 for the abbreviations of tree species names.

**Table A5.** Confusion matrix derived using all data..

| | | Reference | | | | | | | | | |
|---|---|---|---|---|---|---|---|---|---|---|---|
| | | **PKV** | **OB** | **HB** | **EL** | **PY** | **QL** | **CL** | **BL** | **ACB** | **UA (%)** |
| S2(SP+TX+REP_TM)+S1 + Env classification | PKV | 987 | 55 | 8 | 128 | 5 | 105 | 23 | 45 | 14 | 74.52 |
| | OB | 19 | 794 | 53 | 16 | 0 | 79 | 24 | 53 | 22 | 70.49 |
| | HB | 2 | 71 | 779 | 1 | 0 | 23 | 7 | 1 | 0 | 88.58 |
| | EL | 50 | 13 | 3 | 622 | 1 | 15 | 4 | 12 | 3 | 91.54 |
| | PY | 13 | 9 | 2 | 5 | 774 | 33 | 42 | 0 | 47 | 83.75 |
| | QL | 51 | 57 | 3 | 18 | 5 | 371 | 7 | 28 | 32 | 57.92 |
| | CL | 5 | 19 | 5 | 6 | 11 | 29 | 477 | 23 | 23 | 81.37 |
| | BL | 22 | 28 | 1 | 22 | 0 | 14 | 15 | 332 | 10 | 70.08 |
| | ACB | 4 | 20 | 1 | 6 | 25 | 19 | 11 | 5 | 287 | 65.12 |
| PA (%) | | 85.60 | 74.48 | 91.11 | 75.49 | 94.28 | 53.92 | 78.20 | 66.53 | 65.53 | |
| | | OA: 77.98% | | | Kappa: 0.75 | | | | | | |

Note: UA: user's accuracy, PA: producer's accuracy, OA: overall accuracy; see Section 2.2 for the abbreviations of tree species names.

**Table A6.** The selected features use the relevance hierarchical clustering method.

| Cases | Selected Features | Number of Features (after vs. before) |
|---|---|---|
| S2(SP+TX+REP_TM) | 'NDTI', 'NDSVI', 'REP', 'b_17_REP', 'b_4_REP', 'LSWI', 'B5_contrast', 'b_29_REP', 'MTCI', 'b_36_REP', 'B5_corr', 'IRECI', 'b_0_REP', 'B2', 'B4', 'TVI', 'b_19_REP', 'b_31_REP', 'B12', 'b_23_REP', 'B5', 'b_3_REP', 'b_34_REP', 'b_1_REP', 'NDWI', 'b_2_REP', 'B6', 'b_26_REP', 'NDRE1', 'b_13_REP', 'B5_shade', 'B3'. | 32/69 |
| S2(SP+TX+REP_TM) + S1 | 'NDTI', 'NDSVI', 'REP', 'B2', 'b_18_REP', 'VV', 'LSWI', 'b_3_REP', 'b_22_REP', 'NDRE1', 'b_5_REP', 'B5_shade', 'NDI', 'B5_contrast', 'VH', 'b_0_REP', 'B12', 'MTCI', 'b_31_REP', 'B5_corr', 'b_12_REP', 'B5_savg', 'IRECI', 'B4', 'MCARI2', 'b_36_REP', 'TVI', 'B3', 'b_34_REP', 'b_29_REP', 'B6'. | 31/75 |

**Table A6.** *Cont.*

| Cases | Selected Features | Number of Features (after vs. before) |
|---|---|---|
| S2$_{(SP+TX+REP\_TM)}$ + Env | 'NDTI', 'LST_5', 'elevation', 'LST_7', 'NDSVI', 'dec', 'LST_4', 'LSWI', 'REP', 'may', 'LST_9', 'b_5_REP', 'B2', 'jul', 'B5_corr', 'aspect', 'TVI', 'b_1_REP', 'LST_11', 'B11', 'jun', 'B5_contrast', 'b_11_REP', 'MTCI', 'LST_8', 'MCARI2', 'b_0_REP', 'feb', 'b_3_REP', 'NDRE1', 'nov', 'b_15_REP', 'apr', 'B5_shade', 'b_20_REP' | 32/96 |
| S2$_{(SP+TX+REP\_TM)}$ + S1 + Env | 'NDTI', 'elevation', 'LST_7', 'LST_9', 'LST_5', 'NDSVI', 'dec', 'LST_8', 'LST_1', 'NBR', 'oct', 'REP', 'LST_6', 'TVI', 'VV', 'apr', 'may', 'b_1_REP', 'jul', 'b_27_REP', 'b_2_REP', 'b_7_REP', 'b_13_REP', 'b_3_REP', 'b_18_REP', 'b_36_REP', 'B4', 'b_23_REP', 'B2', 'B5', 'B5_contrast', 'NDRE1', 'B5_shade', 'b_25_REP', 'b_0_REP', 'NDWI', 'B12', 'b_10_REP', 'mRVI'. | 32/102 |

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
