# Peer review of "Synergism of Multi-Modal Data for Mapping Tree Species Distribution—A Case Study from a Mountainous Forest in Southwest China"

_remotesensing, doi:10.3390/rs15040979_

Round 1
Reviewer 1 Report
The authors explored multiple satellite images to classify forest species in Southwest China. Specifically, they mainly used the Sentinel bands and SAR retrievals and explored several ML algorithms to identify tree species. The final accuracy is promising, and the study is well-organized. I have several concerns as follows:
1. What’s the typical geolocation error for Sentinel? This is very important to validate your inventory data.
2. In the introduction part, Hyperspectral images have been explored to classify species, be sure to mention relevant references.
3. The introduction is a little long. I suggest the authors can remove some details about background.
4. The manuscript needs moderate English edits. There are some obvious grammar errors throughout the paper.
5. Figure 5 is hard to interpret. Maybe a confusion matrix is easy to understand? Same for Figure 7.
Author Response
Response to Reviewer 1 Comments
Thank you for the reviewers’ comments concerning our manuscript. We have tried our best to improve the manuscript and hope that the correction will meet with approval, the responses to the reviewer’s comments are as flowing:
Comments and Suggestions for Authors:
The authors explored multiple satellite images to classify forest species in Southwest China. Specifically, they mainly used the Sentinel bands and SAR retrievals and explored several ML algorithms to identify tree species. The final accuracy is promising, and the study is well-organized. I have several concerns as follows:
Point 1: What’s the typical geolocation error for Sentinel? This is very important to validate your inventory data.
Response 1: Thank you very much for this question. The geolocation error of Sentinel does exist. As reported by the European Space Agency, the Sentinel-2A products generated between orbits 6003 (acquired on 15 August 2016) and 6011 (acquired on 16 August 2016) suffer important geolocation error from 30m up to 130m as a direct consequence of the collision avoidance maneuver that took place at this time. Fortunately, after carefully checking the Sentinel-2 Anomaly Database, we have found no geolocation anomalies in the products used in this study.
Another issue is about the match accuracy of inventory data and image pixels. We have used many strategies to ensure the match of inventory data and image pixels. The first is that Sentinel-2 Level-1C product, which is composed of 100x100 km2 ortho-tiles, was employed. The overall geolocation accuracy is better than 11 or 12 meters, for about 97 % of the cases, which is about the size of one Sentinel-2 pixel (source from https://sentinel.esa.int/documents/247904/685211/ Sentinel-2_L1C_Data_Quality_Report). Such geolocation errors are acceptable for large scale range of classification tasks. Secondly, we converted the ground survey data into the same UTM/WGS84 projection as the images. Thirdly, the geometric centroid of the pure forest sub-compartment and sub-compartment dominated by one single tree species were used as the spatial position of the sample point, effectively avoiding the problem of inaccurate samples due to geographical errors. In addition, we also visually inspected all sample points and removed abnormal sample points. We tried to ensure the accuracy of the ground validation samples as much as possible.
Point 2: In the introduction part, Hyperspectral images have been explored to classify species, be sure to mention relevant references.
Response 2: Thank you for your good comments. we have modified this part and removed incorrectly phrased sentences. (see lines: 56-59)
Point 3: The introduction is a little long. I suggest the authors can remove some details about background.
Response 3: We have thoroughly checked the whole section and removed some redundant sentences. (see lines: 43-45, 52-54, 75-77, 99-102)
Point 4: The manuscript needs moderate English edits. There are some obvious grammar errors throughout the paper.
Response 4: Thank you for your suggestion. We are sorry for the poor English that caused you difficulties in reading. We have ordered the American Journal Experts (AJE), a professional agency offering English editing, to polish our paper. We provided editorial certificates.
Point 5: Figure 5 is hard to interpret. Maybe a confusion matrix is easy to understand? Same for Figure7.
Response 5: Thank you very much, as suggested we replaced the confusion matrix (Table A5) with Figure 5. We have also updated Figure 7 (now Figure 6) for better visualization.

Reviewer 2 Report
The innovation of the manuscript is relatively common, and the conclusion needs to be further summarized.
(1) The Latin names of tree species should be written normal.
(2) Many different VIs have been used in this study, however, the resolution of different bands in Sentinel-2 data are inconsistent. How to solve this problem and how to ensure that the characteristics of tree species can be reflected in the coarse resolution (below 10m)? In addition, synthetic images were adopted in VIs calculation, how to ensure consistency in time for different pixels? It means that the great growth differences in time between each pixel should be considered.
(3) The spatial resolution of Rededge1 is 20m, and the spatial resolution of its texture feature will be further coarser. On this scale, it is hard to discuss the identifiability of tree species.
(4) At the beginning, any additional features should have its own potential application significance. The author introduced a large number of features in the 2.2.1, but whether all of them are necessary needs further consideration of their potential classification ability in different tree species, especially the VIs.
(5) How to match the spatial resolution data of each feature consistently, some are very coarse, some are detailed.
(6) The results are not credible enough. Especially in the feature selection part, the author also mentioned that the selected features are not very good by applying the J-M. In addition, the significance of remote sensing information is lower than environmental factors, which is insufficient in this study.
(7) Results showed that “The best performance, with overall accuracy 77.98%, was achieved from the scenario related to all features and the random forest classifier.” is not obvious(Figure 4) . Maybe one or two validation points could change this.
(8) It is mentioned that “Mountainous Forest” in the title? However, the features and methods do not focus on this, and thus the innovation is not obvious.
Author Response
Response to Reviewer 1 Comments
Thank you for the reviewers’ comments concerning our manuscript. We have tried our best to improve the manuscript and hope that the correction will meet with approval, the responds to the reviewer’s comments are as flowing:
The innovation of the manuscript is relatively common, and the conclusion needs to be further summarized.
Response: Thank you very much for this candid comment. We are very sorry that we have not explained our innovation clearly. Serious cloud contamination and complex vegetation composition are main hindrance for classifying tree species in large mountainous areas. The optical images are constrained by the cloud contamination, and the addition of auxiliary data, especially environmental factors related to their distribution, may alleviate such problems. The motivation of this study is to explore the effectiveness of synergism of multi-modal data in a typical mountainous forest area. We design different feature combinations to evaluate the contribution of different features to discriminate tree species. In this process, we filtered and ranked the different features through correlation hierarchical clustering. The results can provide a reference feature set for future large-scale mountainous tree species classification. To address this innovation, we have revised the conclusion section.
Point 1: The Latin names of tree species should be written normal.
Response 1: We have checked and modified the Latin names of tree species in the full text. We have revised the Latin name of Simao pine to (Pinus kesiya var. langbianensis) (see lines: 185), fir's Latin name is revised to (Cunninghamia lanceolata (Lamb.) Hook). (see lines:187), birch's Latin name is revised to (Betula) (see lines:187). The same revision was made in lines 299-302.
Point 2: Many different VIs have been used in this study, however, the resolution of different bands in Sentinel-2 data are inconsistent. How to solve this problem and how to ensure that the characteristics of tree species can be reflected in the coarse resolution (below 10m)?
Response 2: Thank you for these insightful comments. As the resolution of different bands in Sentinel-2 data are inconsistent, we firstly used the GEE function (resample) to resample all bands of Sentinel-2 to 10m. It is true that the average tree crown area of most tree species in our research area is smaller than 100 m2. This fact means a single pixel reflects a mixed signal of several tree crowns. From this view, our pixel-based classification aims to classifying stand tree species, rather than a single tree. To achieve this goal, the training samples were generally located as the geometric centroid of the pure forest sub-compartments and sub-compartments dominated by one single tree species and the areas of sub-compartments are far larger than the coverage area of a given pixel. These strategies may ensure that a reference pixel reflect a mixed characteristics of several trees belonging to the same species.
Point 3: In addition, synthetic images were adopted in VIs calculation, how to ensure consistency in time for different pixels? It means that the great growth differences in time between each pixel should be considered.
Response 3: Thank you for this question. The temporal inconsistency of pixels truly exists in the synthetic images. The main reason caused this problem is that we cannot synthesize qualified images only with image tiles acquired in year 2016. The composite images for January, May, and September 2016 (Figure 1) suffers from serious cloud contamination. Meanwhile, it is also difficult to obtain a cloud-free image even with the synthesis of one year of observations.
- The composite images from different time range. (a)-(c) represent the median composites over the entire study area form January, May, and September 2016, respectively. (d)-(f) represent the median composites of the partial region for April to October 2016, 2016, and 2015-2017.
In addition, we show the spectral reflectance of a tree species with different methods of synthesis. As shown in the figure 2 below, the spectral variation of the different synthetic methods is not significant for the tree species. We found that the reflectance differences with different synthesis methods are slight in all bands. In addition, different tree species have highly similar spectral curves. The high quality images also ensure the VIs credibility. Such a synthetic method truely sacrifices the information of the VIs in time. Therefore, we generated a time series of vegetation indices to capture the growth differences in times among tree species.
- The spectral reflectance of tree species with different methods of synthesis.
Point 4: The spatial resolution of Rededge1 is 20m, and the spatial resolution of its texture feature will be further coarser. On this scale, it is hard to discuss the identifiability of tree species.
Response 4: Thank you for your comment. According to our experimental results, this operation truly does not add any useful information. The initial choice on the Rededge1 band is based on the common sense that the red-edge band is sensitive to capture imperceptible differences between vegetation. And we expected the texture calculation can capture more macroscopic spatial patterns, which may be related to tree stand structures. Unfortunately, our experimental results found that the texture features calculated at the 10m scale did not significantly improve the results. we agree with the reviewer’s point and ascribe this result to the low spatial resolution of the original band. We also discussed this in the paper. (Please see lines: 560-570)
Point 5: At the beginning, any additional features should have its own potential application significance. The author introduced a large number of features in the 2.2.1, but whether all of them are necessary needs further consideration of their potential classification ability in different tree species, especially the VIs.
Response 5: Thank you for your comment. The Sentinel 2 base band is necessary and in addition, we did use a large number of vegetation indices, including spectral and radar vegetation indices. These vegetation indices are introduced from previous works (Table 1 for details).
And these indices are related to the growth state of vegetation as well as physiological characteristics, but it was not explored whether these indices are useful for tree species classification. Therefore, we collected a large feature pool in the 2.2.1. Thereafter, based on the results of feature importance evaluation and feature selection, we found that some spectral vegetation indices were significant contributors to tree species classification in this scene, while the radar indices were not as effective, which can provide a reference for future tree species classification in similar scenes. We also added the explanation in section 4.2. (Please see lines: 552-556)
- The vegetation indices and corresponding formulas were used in this study.
|
Indices |
Formulation |
Reference |
|
TVI |
[1] |
|
|
MTCI |
[2] |
|
|
NBR |
[3] |
|
|
IRECI |
[4] |
|
|
MCARI2 |
[5] |
|
|
LSWI |
[6] |
|
|
NDRE |
[7] |
|
|
Chl NDI |
[8] |
|
|
REP |
[9] |
|
|
NDVI |
[10] |
|
|
NDSVI |
[11] |
|
|
NDTI |
[12] |
|
|
NDSI |
[13] |
|
|
NDBI |
[14] |
|
|
NDWI |
[15] |
|
|
CIrededge |
[16] |
Point 6: How to match the spatial resolution data of each feature consistently, some are very coarse, some are detailed.
Response 6: Thank you for your important questions. All our data is derived from the Google Earth Engine platform and are also resampled to 10m using a bilinear interpolation operator under the platform. Although the resolution of environmental data is 1 km, they have little change in small areas and can be used as auxiliary data. Our results also verify that such environmental factors are important for improving tree species classification, due to their direct correlation with the distribution of tree species.
Point 7: The results are not credible enough. Especially in the feature selection part, the author also mentioned that the selected features are not very good by applying the J-M. In addition, the significance of remote sensing information is lower than environmental factors, which is insufficient in this study.
Response 7: Thank you for your insightful comment. We apologize for the unclear explanation in the feature selection part. We used the correlation hierarchical clustering method for feature selection, and J-M distance is used to quantitatively evaluate the feature importance for different feature combinations. And the J-M distances were calculated based on 1000 randomly selected samples of each tree species. Based on these samples, Figure 5 in the manuscript indicates the significance of Sentinel-2 is slightly lower than environmental factors. If the selected samples are presentative enough, the feature sets with higher separability performance should have better performance on classification. However, another factor that should be considered in this case is that the environmental factors are with lower spatial resolution 1km. Although those factors were unsampled to10m, no extra spatial detail was added. This conclusion is also confirmed by Figure 8 and 9. The prediction result using environmental data (Figure 8-(e), Figure 9-(e)) has much less much information compared to that using Sentinel-2. We add a description of this in Section 3.2 (see lines 452-453).
Figure.5 (source in the manuscript) Class separability results derived based on the JM distance: (a) - (f) represent case 1- case 7.
Figure.9 (source in the manuscript) Detailed view of the tree-species maps with worse interpretation derived under different cases. (a) false-color image synthesized from the green, blue, and NIR bands; (b) ground survey labels; and (c)~(i) RF maps predicted based on the 7 feature sets.
Point 8: Results showed that “The best performance, with overall accuracy 77.98%, was achieved from the scenario related to all features and the random forest classifier.” is not obvious (Figure 4). Maybe one or two validation points could change this.
Response 8: Thank you for this suggestion. As suggested we updated Figure 4 in the manuscript. Also, Figure 8-(i), Figure 9-(i) in the manuscript show good classification results for all features on random forest for some areas.
In addition, we repeat the classification process for all features on the random forest model 10 times, with randomly selected training and validation samples each time. The results are as follows:
|
Number |
1 |
2 |
3 |
4 |
5 |
6 |
7 |
8 |
9 |
10 |
|
Accuracy |
77.98% |
78.15% |
77.27% |
77.46% |
77.54% |
77% |
77.85% |
77.28% |
77.72% |
77.54% |
The average precision of the 10 experiments was 77.58 and the variance was 0.1117. This also verifies the robustness and reliability of our experimental results.
Point 9: It is mentioned that “Mountainous Forest” in the title? However, the features and methods do not focus on this, and thus the innovation is not obvious.
Response 9: Thank you for your question. We are very sorry that we did not make the innovation clear. Our study area is a typical mountainous terrain, in the subtropical and tropical regions, with a complex forest system structure, irregular topography, and serious cloud contamination, making optical imagery limited for use on a large scale. Also, the study area has diverse climatic and topography characteristics, and our intention was to explore synergistic multimodal data to improve tree species classification in mountainous areas.
The cited references to this comment are listed as follows:
[1] Main, R., Cho, M.A., Mathieu, R., O’Kennedy, M.M., Ramoelo, A., & Koch, S. (2011). An investigation into robust spectral indices for leaf chlorophyll estimation. ISPRS Journal of Photogrammetry and Remote Sensing, 66, 751-761.
[2] Dash, J., & Curran, P. (2004). The MERIS terrestrial chlorophyll index, 25, 5403-5413.
[3] Long, T., Zhang, Z., He, G., Jiao, W., Tang, C., Wu, B., Zhang, X., Wang, G., & Yin, R. (2019). 30 m resolution global annual burned area mapping based on Landsat Images and Google Earth Engine. Remote Sensing, 11, 489.
[4] Rozenstein, O., Haymann, N., Kaplan, G., & Tanny, J. (2019). Validation of the cotton crop coefficient estimation model based on Sentinel-2 imagery and eddy covariance measurements. Agricultural Water Management, 223, 105715.
[5] Wu, C., Niu, Z., Tang, Q., & Huang, W. (2008). Estimating chlorophyll content from hyperspectral vegetation indices: Modeling and validation. Agricultural and Forest Meteorology, 148, 1230-1241.
[6] Chandrasekar, K., Sesha Sai, M., Roy, P., & Dwevedi, R. (2010). Land Surface Water Index (LSWI) response to rainfall and NDVI using the MODIS Vegetation Index product. International Journal of Remote Sensing, 31, 3987-4005.
[7] Ahamed, T., Tian, L., Zhang, Y., & Ting, K. (2011). A review of remote sensing methods for biomass feedstock production. Biomass and bioenergy, 35, 2455-2469.
[8] Richardson, A.D., Duigan, S.P., & Berlyn, G.P. (2002). An evaluation of noninvasive methods to estimate foliar chlorophyll content. New phytologist, 153, 185-194.
[9] Schlerf, M., Atzberger, C., & Hill, J. (2005). Remote sensing of forest biophysical variables using HyMap imaging spectrometer data. Remote Sensing of Environment, 95, 177-194.
[10] Carlson, T.N., & Ripley, D.A. (1997). On the relation between NDVI, fractional vegetation cover, and leaf area index. Remote Sensing of Environment, 62, 241-252.
[11] Zheng, G., Bao, A., Li, X., Jiang, L., Chang, C., Chen, T., & Gao, Z. (2019). The Potential of Multispectral Vegetation Indices Feature Space for Quantitatively Estimating the Photosynthetic, Non-Photosynthetic Vegetation and Bare Soil Fractions in Northern China. Photogrammetric Engineering & Remote Sensing, 85, 65-76.
[12] Zhong, L., Hu, L., & Zhou, H. (2019). Deep learning based multi-temporal crop classification. Remote Sensing of Environment, 221, 430-443.
[13] Richardson, A.D., Duigan, S.P., & Berlyn, G.P. (2002). An evaluation of noninvasive methods to estimate foliar chlorophyll content. New phytologist, 153, 185-194.
[14] Zhang, Y., Odeh, I.O., & Han, C. (2009). Bi-temporal characterization of land surface temperature in relation to impervious surface area, NDVI and NDBI, using a sub-pixel image analysis. International Journal of Applied Earth Observation and Geoinformation, 11, 256-264.
[15] Gao, B.-C. (1996). NDWI—A normalized difference water index for remote sensing of vegetation liquid water from space. Remote Sensing of Environment, 58, 257-266.
[16] Duan, B., Fang, S., Gong, Y., Peng, Y., Wu, X., & Zhu, R. (2021). Remote estimation of grain yield based on UAV data in different rice cultivars under contrasting climatic zone. Field Crops Research, 267, 108148.
Reviewer 3 Report
A breief summary
Mapping tree species using remote sensing data is a challenging task that is currently at the research front. There are long list of methods and datasets applied to mapping tree species by different researchers in different regions of the world. Seen in that regard, the article is scientifically relevant. The manuscript addresses the challenges of mapping tree species in tropical and subtropical regions using remote sensing and environmental data. It particularly contributes in how environmental and remote sensing data could be combined to map tree species in tropical and subtropical regions. Seven different sets of data are put together and three different classification methods are tested. Although the intentions/objectives are clear, the research design has flaws as will be commented hereunder.
The manuscript has the potential for publication if major review is undertaken. I suggest the following areas of improvement if they decide to make major review. I will keep specific comments for the revised version as major changes are expected.
General concept comments
1. 1. The language has to be improved considerably. I advice an assistance by a native English speaker or a professional English editor.
2. The title itself is confusing and should be changed to a more concise one.
3. Seven different sets of data are compared. (The word scenario is used to describe these cases. Scenario implies a possible course of action or events. I advice the use of other words such as case or dataset or even sets of data, etc. ). The inclusion of some of the obvious cases make the scientific design of the work . For example, the attempt to map 9 different tree species using only radar data is a waste of time. The research should have been designed in such a way that a baseline method and dataset is defined and a proposed method (alternative methods) should have been compared to that. The design of the seven sets of data has to therefore be revisited and any choice of datasets has to be justified in reference to previous works.
4. Three classification methods, namely RF, SVM and XGBoost are also compared. What is the reason for the need of the comparison? Each of these have their own hyperparameters. There is a need for systematic tuning of the hyperparameters to achieve the potential accuracies of the algorithms. Perhaps the difference between untuned algorithm and a tuned version of the same algorithm might surpass the difference between the three algorithms.
5. While existing works show that one tree species mapping benefits from Sentinel-2 time series, only the texture data is in time series. It is not clear why this choice is made.
6. Presentations of the results are confusing, to say the least. Specifically, figures 5 and 7 (also many others) do not make anything clearer. I think the presentation of the result suffers from the side effects of using unnecessarily too many sets of data and methods (7 cases of datasets and 3 methods).
7. At times, the manuscript gives the impression of an ecological study rather than a geographical as the presentation of the results took detour to topics like the effect of topography on tree species distribution, the effect of precipitation on tree species distribution, etc. This diverts the focus from the main objective. The discussion even looses track and takes on new major topics.
8. The manuscript should be reduced significantly and focused more on the main objective.
9. Please take care of all spelling and other syntax errors before submitting the manuscript again.
Author Response
Response to Reviewer 1 Comments
Thank you for the reviewers’ comments concerning our manuscript. We have tried our best to improve the manuscript and hope that the correction will meet with approval, the responds to the reviewer’s comments are as flowing:
A brief summary
Mapping tree species using remote sensing data is a challenging task that is currently at the research front. There are long list of methods and datasets applied to mapping tree species by different researchers in different regions of the world. Seen in that regard, the article is scientifically relevant. The manuscript addresses the challenges of mapping tree species in tropical and subtropical regions using remote sensing and environmental data. It particularly contributes in how environmental and remote sensing data could be combined to map tree species in tropical and subtropical regions. Seven different sets of data are put together and three different classification methods are tested. Although the intentions/objectives are clear, the research design has flaws as will be commented hereunder.
The manuscript has the potential for publication if major review is undertaken. I suggest the following areas of improvement if they decide to make major review. I will keep specific comments for the revised version as major changes are expected.
General concept comments
Point 1: The language has to be improved considerably. I advise an assistance by a native English speaker or a professional English editor.
Response 1: Thank you for your suggestion. We are sorry that the English grammar errors have caused you difficulties in reading. We have asked American Journal Experts (AJE), a professional agency offering English editing, to polish our paper. We provided editorial certificates.
Point 2: The title itself is confusing and should be changed to a more concise one.
Response 2: After discussion, we have replaced the title with a more concise one: Synergism of Multi-modal Data for Mapping Tree Species Distribution. A Case Study from a Mountainous Forest in Southwest China.
Point 3: Seven different sets of data are compared. (The word scenario is used to describe these cases. Scenario implies a possible course of action or events. I advise the use of other words such as case or dataset or even sets of data, etc.). The inclusion of some of the obvious cases make the scientific design of the work. For example, the attempt to map 9 different tree species using only radar data is a waste of time. The research should have been designed in such a way that a baseline method and dataset is defined and a proposed method (alternative methods) should have been compared to that. The design of the seven sets of data has to therefore be revisited and any choice of datasets has to be justified in reference to previous works.
Response 3: As suggested we replace the full scenario with the term “case”. A previous review (Review of studies on tree species classification from remotely sensed data) indicated that Most studies were conducted in temperate and boreal forest ecosystems of North America and Europe while the number of studies conducted in Asia, Africa, South America and Australia was sparse and not developed to account for a large variety of ecological conditions [1]. And the same data and methods may yield inconsistent conclusions in different application scenarios. In a study for tree species classification in a Central European Biosphere Reserve, the overall accuracy (OA) of 55.7% was obtained using only sentinel-1 data. Our intention was also to explore whether sentinel-1 could be useful in the classification of tree species in mountainous area. However, the results showed that it is difficult to effectively distinguish tree species due to the complex topography and high forest density in mountainous forests.
Point 4: Three classification methods, namely RF, SVM and XGBoost are also compared. What is the reason for the need of the comparison? Each of these have their own hyperparameters. There is a need for systematic tuning of the hyperparameters to achieve the potential accuracies of the algorithms. Perhaps the difference between untuned algorithm and a tuned version of the same algorithm might surpass the difference between the three algorithms.
Response 4: Thank you for your suggestion. We chose the three classifiers due to the following reasons: (1) The three machine learning classifiers were widely accepted as advanced classifiers for vegetation related studies [2,3]. (2) In addition, we wanted to verify if the selected features can give credible results in different classification models. For this purpose, the description is given in 643-656.
In this experiment, for the random forest, we adjusted the numberOfTrees and tried different settings of 100, 120, 150, 180 according to some previous studies and the number of features used in this study, with variablesPerSplit using the default of null. For the SVM, we chose 'RBF' as the kernel function, the gamma and cost were also manually adjusted several times. For the XGBoost, we manually adjusted the numberOfTrees and shrinkage parameters, and the samplingRate was set to the default value of 0.7.
Point 5: While existing works show that one tree species mapping benefits from Sentinel-2 time series, only the texture data is in time series. It is not clear why this choice is made.
Response 5: Thank you for the suggestion. Sentinel-2 time series was generated based on the REP vegetation index, and the texture features were calculated for the rededge band. We improved the description of the time series generation. (see lines: 237)
Point 6: Presentations of the results are confusing, to say the least. Specifically, figures 5 and 7 (also many others) do not make anything clearer. I think the presentation of the result suffers from the side effects of using unnecessarily too many sets of data and methods (7 cases of datasets and 3 methods).
Response 6: Thank you very much for your constructive suggestions. As suggested we replaced the confusion matrix (Table A5) with Figure 5. We have also updated Figure 7 (now Figure 6) in a more visual way.
Point 7: At times, the manuscript gives the impression of an ecological study rather than a geographical as the presentation of the results took detour to topics like the effect of topography on tree species distribution, the effect of precipitation on tree species distribution, etc. This diverts the focus from the main objective. The discussion even loses track and takes on new major topics.
Response 7: Thank you very much for your question. It is a very insightful comment. Our intention was to explore synergistic multimodal data on the role of tree species classification in mountainous areas. In the discussion section, we emphasized the importance of environmental factors because we wanted to verify the reliability of our results by analyzing the correlation between the ecological conditions of tree species and their distribution. We also referred to a large literature and found that few studies have paid attention to the environmental factors on tree species classification. Our results also reflect a geographical cognitive consistency with the distribution of tree species. This also confirms that our method has an improved effect on the tree species classification in mountainous areas.
Point 8: The manuscript should be reduced significantly and focused more on the main objective.
Response8: Thank you very much for your positive comments and constructive suggestions. As suggested we improved the introduction and removed some redundant information to make it concise. (see lines: 43-45, 52-54, 75-77, 99-102). We have removed some common knowledge introduction to classifiers in section 2.4.1, such as the process of implementing the XGBoost algorithm. (see lines: 341-342, 363-348) We revisited the discussion section, removed some discussions that seemed to have little relevance to the topic, and focused on the main ideas.
Point 9: Please take care of all spelling and other syntax errors before submitting the manuscript again.
Response 9: Thank you for your suggestion. We are sorry that the English grammar errors have caused you difficulties in reading. We have asked American Journal Experts (AJE), a professional agency offering English editing, to polish our paper. After that, we also carefully revised and checked the revised manuscript to ensure the elimination of spelling and other syntax errors.
The cited references to this comment are listed as follows:
[1] Fassnacht, F.E.; Latifi, H.; Stereńczak, K.; Modzelewska, A.; Lefsky, M.; Waser, L.T.; Straub, C.; Ghosh, A. Review of studies on tree species classification from remotely sensed data. Remote Sensing of Environment 2016, 186, 64-87, doi:10.1016/j.rse.2016.08.013.
[2] Wessel, M.; Brandmeier, M.; Tiede, D. Evaluation of Different Machine Learning Algorithms for Scalable Classification of Tree Types and Tree Species Based on Sentinel-2 Data. Remote Sensing 2018, 10, doi:10.3390/rs10091419.
[3] You, H.; Huang, Y.; Qin, Z.; Chen, J.; Liu, Y. Forest Tree Species Classification Based on Sentinel-2 Images and Auxiliary Data. Forests 2022, 13, doi:10.3390/f13091416.

Round 2
Reviewer 3 Report
I have attached my comments.

Author Response
Response to Reviewer 3 Comments
Dear reviewers:
Thank you for your comments concerning our manuscript. I really appreciate your intensive work, especially for your specific comments, it is significant for the improvement of the paper. We have read through comments carefully and have made corrections. Based on the instructions provided in your comments, we uploaded the file of the revised manuscript. Revisions in the text are shown using red highlight for additions, and strikethrough font for deletions. The responses to the reviewer's comments are marked yellow and used annotation function of MS word and presented following:
General concept comments
Point a: The rebuttal given to my comments are ought to be included in the manuscript and referred to with line number (should not be relied to me). Some rebuttal comments are not included in the manuscript.
Response a: Thank you for your suggestion. We are sorry that we did not display the rebuttal comments clearly. We included all rebuttal comments and marked all responses to comments the revised manuscript using the annotation function of MS word.
Point b: The research design (7 cases, 3 methods) is still not convincing. Sufficient changes or rebuttal have not been given to my previous comment (point 3). Strong justifications must be given in the introduction.
Response b: Thank you for your suggestion. This design is mainly considered from two aspects:
- We constructed 7 cases including single and multiple combined feature sets to assess their performance for tree species mapping. We did not know these data effects for our study area and for our subjects before the experiment. Even when using the same data, differences in environmental conditions, distribution of species and quality of observations in the study area can affect the classification results. Therefore, we used a design similar to that of previous studies to evaluate these data [1,2,3,4].
- There machine learning methods were used for training and prediction in this study to obtain a reliable evaluation on the contribution of multi-source data. We consider that a single algorithm may not be explainable. The data we used obtained similar performance on different algorithms, indicating that these data is credible.
Based on these two points, we have also strengthened the explanation in the introduction section and list the literature that contains such designs. (line:121-125)
Point c: Most seriously, the discussion section contains new results which are not presented in the result section. Please either combine them under the same title (result and discussion) or distinctly present the results under the result section and only make discussion of the presented results under the discussion section. Major change is required here.
Response c: We are extremely grateful to reviewer for pointing out this problem. We have reorganized the structure of the results and discussion sections. We merged the classification result sections and removed the part of the discussion related to environmental factors and repetition to focus on our main objective. More specifically, following the suggestions of the reviewers, we have removed specific comment on Figure 11 and its description. However, we retained some discussion of the correlation between tree species and environmental factors, and this can be helpful for the interpretation of our classification results.
Point d: Some of the figures are still either confusing or of no additional value: for example, figure 6.
Response d: Thank you for your suggestion. The features in Figure 6 are derived from the best combination of features on RF in this experiment. We evaluate the importance of features based on their scores in the classification process. Figure 6 allows us to understand which features are dominant for the tree species classification in mountainous areas and provides a reference for similar studies in the future. So, we retained feature importance ranking and it’s necessary to explain the magnitude of contribution of the features used in this paper. To make the information in the figure clearer, we have changed the names of some features and added notes to the figure titles. (line: 494-498)
Point e: Despite the claim that a professional has edited the language, there are still numerous flaws with the language as partially notified in the specific comments hereunder.
Response e: We sincerely appreciate your specific comments. These flaws are serious and we have carefully checked and revised according to your valuable suggestions.
Reference:
[1] Liu, Y.; Gong, W.; Xing, Y.; Hu, X.; Gong, J. Estimation of the forest stand mean height and aboveground biomass in North-east China using SAR Sentinel-1B, multispectral Sentinel-2A, and DEM imagery. ISPRS Journal of Photogrammetry and Remote Sensing 2019, 151, 277-289, doi:10.1016/j.isprsjprs.2019.03.016.
[2] Lechner, M.; Dostálová, A.; Hollaus, M.; Atzberger, C.; Immitzer, M. Combination of Sentinel-1 and Sentinel-2 Data for Tree Species Classification in a Central European Biosphere Reserve. Remote Sensing 2022, 14, doi:10.3390/rs14112687.
[3] Xu, P.; Tsendbazar, N.-E.; Herold, M.; Clevers, J.G.P.W.; Li, L. Improving the characterization of global aquatic land cover types using multi-source earth observation data. Remote Sensing of Environment 2022, 278, doi:10.1016/j.rse.2022.113103.
[4] Zhou, W.; Liu, Y.; Ata-Ul-Karim, S.T.; Ge, Q.; Li, X.; Xiao, J. Integrating climate and satellite remote sensing data for predicting county-level wheat yield in China using machine learning methods. International Journal of Applied Earth Observation and Geoinformation 2022, 111, doi:10.1016/j.jag.2022.102861.
We would love to thank you for allowing us to resubmit a revised copy of the manuscript and we highly appreciate your time and consideration.
Sincerely,
Pengfei Zheng.
